# Towards a transferable fermionic neural wavefunction for molecules

Michael Scherbela[1,4], Leon Gerard[2,4] & Philipp Grohs [1,2,3]✉

Deep neural networks have become a highly accurate and powerful wavefunction ansatz in combination with variational Monte Carlo methods for solving the electronic Schrödinger equation. However, despite their success and favorable scaling, these methods are still computationally too costly for wide adoption. A significant obstacle is the requirement to optimize the wavefunction from scratch for each new system, thus requiring long optimization. In this work, we propose a neural network ansatz, which effectively maps uncorrelated, computationally cheap Hartree-Fock orbitals, to correlated, high-accuracy neural network orbitals. This ansatz is inherently capable of learning a single wavefunction across multiple compounds and geometries, as we demonstrate by successfully transferring a wavefunction model pre-trained on smaller fragments to larger compounds. Furthermore, we provide ample experimental evidence to support the idea that extensive pre-training of such a generalized wavefunction model across different compounds and geometries could lead to a foundation wavefunction model. Such a model could yield high-accuracy ab-initio energies using only minimal computational effort for fine-tuning and evaluation of observables.

Accurate predictions of quantum mechanical properties for molecules is of utmost importance for the development of new compounds, such as catalysts, or pharmaceuticals. For each molecule the solution to the Schrödinger equation yields the wavefunction and electron density, and thus in principle gives complete access to its chemical properties. However, due to the curse of dimensionality, computing accurate approximations to the Schrödinger equation quickly becomes computationally intractable with increasing number of particles. Recently, deep-learning-based Variational Monte Carlo (DL-VMC) methods have emerged as a high-accuracy approach with favorable scaling $\mathcal{O}(N^4)$ in the number of particles $N$[1]. These methods use a deep neural network as ansatz for the high-dimensional wavefunction, and minimize the energy of this ansatz to obtain the ground-state wavefunction. Based on two major architectures for the treatment of molecules in first quantization, PauliNet[1] and FermiNet[2], several improvements and applications have emerged. On the one hand, enhancements of architecture, optimization and overall approach have led to substantial

improvements in accuracy or computational cost[3–7]. On the other hand, these methods have been adapted to many different systems and observables: model systems of solids[8,9], real solids[10], energies and properties of individual molecules[1,2,5,11], forces[12,13], excited states[14] and potential energy surfaces[13,15,16]. Furthermore, similar methods have been developed and successfully applied to Hamiltonians in second quantization[17,18].

We want to emphasize that DL-VMC is an ab-initio method, that does not require any input beyond the Hamiltonian, which is defined by the molecular geometry. This differentiates it from surrogate models, which are trained on results from ab-initio methods to either predict wavefunctions[19,20] or observables[21].

Despite the improvements in DL-VMC, it has not yet been widely adopted, in part due to the high computational cost. While DL-VMC offers favorable scaling, the method suffers from a large prefactor, caused by an expensive optimization with potentially slow convergence towards accurate approximations. Furthermore this

[1]Faculty of Mathematics, University of Vienna, Vienna, Austria. [2]Research Network Data Science, University of Vienna, Vienna, Austria. [3]Johann Radon Institute for Computational and Applied Mathematics, Austrian Academy of Sciences, Linz, Austria. [4]These authors contributed equally: Michael Scherbela, Leon Gerard. ✉e-mail: philipp.grohs@univie.ac.at

optimization needs to be repeated for every new system, leading to prohibitively high computational cost for large-scale use. This can be partially overcome by sharing a single ansatz with identical parameters across different geometries of a compound, allowing more efficient computation of Potential Energy Surfaces (PES)[13,15,16]. However, these approaches have been limited to different geometries of a single compound and do not allow successful transfer to new compounds. A key reason for this limitation is that current architectures explicitly depend on the number of orbitals (and thus electrons) in a molecule. Besides potential generalization issues, this prevents a transfer of weights between different compounds already by the fact that the shapes of weight matrices are different for compounds of different size.

In this work we propose a neural network ansatz, which does not depend explicitly on the number of particles, allowing to optimize wavefunctions across multiple non-periodic, gas-phase compounds with multiple different geometric conformations. We find, that our model exhibits strong generalization when transferring weights from small molecules to larger, similar molecules. In particular we find that our method achieves high accuracy for the important task of relative energies. Our approach is inspired by the success of foundation models in language[22] or vision[23,24], where models are extensively pre-trained and then applied to new tasks−either without any subsequent training (referred to as zero-shot evaluation) or after small amount of training on the new task (referred to as fine-tuning). Zhang et al.[25] have shown that this paradigm can be successfully applied to wavefunctions, in their case for model Hamiltonians in second quantization.

In this work, we pre-train a first base-model for neural network wavefunctions in first quantization, and evaluate the pre-trained model by performing predictions on chemically similar molecules (in-distribution) and disparate molecules (out-of-distribution). We find that our ansatz outperforms conventional high-accuracy methods such as CCSD(T)-ccpVTZ and that fine-tuning our pre-trained model reaches this accuracy ≈ 20x faster, than optimizing a new model from scratch. When analyzing the accuracy as a function of pre-training resources, we find that results systematically and substantially improve by scaling up either the model size, data size or number of pre-training steps. These results could pave the way towards a foundation wavefunction model, to obtain high-accuracy ab-initio results of quantum mechanical properties using only minimal computational effort for fine-tuning and evaluation of observables.

Additionally we compare our results to GLOBE, a concurrent work[26], which proposes reparameterization of the wavefunction based on machine-learned, localized molecular orbitals. We find that for the investigated setting of re-using pre-trained weights our method in comparison achieves lower (and thus more accurate) absolute energies, higher accuracy of relative energies and is better able to generalize across chemically different compounds.

We use the following notation throughout this work: All vectors, matrices and tensors are denoted in **bold letters**, including functions with vectorial output. The $i$-th electron position for $i \in \{1, ..., n_{el}\}$ is denoted as $\mathbf{r}_i \in \mathbb{R}^3$. The set $\{\mathbf{r}_1, ..., \mathbf{r}_{n_\uparrow}\}$ of all electrons with spin up is abbreviated with $\{\mathbf{r}_\uparrow\}$, the set $\{\mathbf{r}_{n_\uparrow+1}, ..., \mathbf{r}_{n_{el}}\}$ of all spin-down electrons as $\{\mathbf{r}_\downarrow\}$. Similarly, the nuclear positions and charges of a molecule are denoted by $\mathbf{R}_I \in \mathbb{R}^3$ and $Z_I \in \mathbb{N}$, $I = 1, ..., N_{atoms}$, with the set of all nuclear positions and their corresponding charges denoted as $\{(\mathbf{R}, \mathbf{Z})\}$. Indices $i, k \in \{1, ..., n_{el}\}$ correspond to electrons and orbitals respectively, whereas $I, J \in \{1, ..., N_{atoms}\}$ correspond to nuclei. By $\langle \cdot, \cdot \rangle$ the dot product is denoted.

## Results

In the following, we briefly outline our approach and how it extends existing work (A multi-compound wavefunction ansatz). We show the fundamental properties of our ansatz such as extensivity and equivariance with respect to the sign of reference orbitals. We demonstrate

the transferability of the ansatz when pre-training on small molecules and re-using it on larger, chemically similar molecules. We also compare its performance against GLOBE, a concurrent work[26]. Lastly, we present a first wavefunction base model pre-trained on a large diverse dataset of 360 geometries and evaluate its downstream performance.

## A multi-compound wavefunction ansatz

Existing high-accuracy ansätze for neural network wavefunctions $\psi$ all exhibit the following structure:

$$\mathbf{h}_i = \mathbf{h}_\theta(\mathbf{r}_i, \{\mathbf{r}_\uparrow\}, \{\mathbf{r}_\downarrow\}, \{(\mathbf{R}, \mathbf{Z})\})$$
$$\mathbf{h}_i \in \mathbb{R}^{D_{emb}}, \quad i = 1 \dots n_{el} \tag{1}$$

$$\Phi_{ik}^d = \varphi_k^d(\mathbf{r}_i) \langle \mathbf{F}_k^d, \mathbf{h}_i \rangle$$
$$\varphi_k^d(\mathbf{r}_i) : \mathbb{R}^3 \to \mathbb{R}, \quad \mathbf{F}_k^d \in \mathbb{R}^{D_{emb}}$$
$$i, k = 1 \dots n_{el}, \quad d = 1 \dots N_{det} \tag{2}$$

$$\psi = \sum_{d=1}^{N_{det}} \det \left[ \Phi_{ik}^d \right]_{i,k=1\dots n_{el}} \tag{3}$$

The neural network $\mathbf{h}_\theta$ in eq. (1) computes a $D_{emb}$-dimensional embedding $\mathbf{h}_i$ of electron $i$, by taking in information of all other particles, e.g., by using attention or message passing. Eq. (2) maps these high-dimensional embeddings onto $n_{el} \times N_{det}$ orbitals (indexed by $k$), using trainable backflow matrices $\mathbf{F}^d$ and typically trainable envelope functions $\varphi_k^d : \mathbb{R}^3 \to \mathbb{R}$. Eq. (3) evaluates the final wavefunction $\psi$ as a sum of (Slater-)determinants of these orbitals, to ensure antisymmetry with respect to permutation of electrons.

By considering the interaction of all particles, in particular with the sets $\{\mathbf{r}_\uparrow\}$ and $\{\mathbf{r}_\downarrow\}$, the functions in Eq. (2) account for the inter-particle correlation and therefore are able to better represent the true ground-state wavefunction. If the orbitals $\Phi_{ik}^d$ would only depend on $\mathbf{r}_i$ (instead of the many-body embedding $\mathbf{h}_i$), they would correspond to single-particle functions, e.g. Hartree-Fock orbitals. Existing methods, as proposed in Pfau et al.[2] or Hermann et al.[1], differ in the way the embedding $\mathbf{h}_i$ and the envelope functions $\varphi_k^d$ are built. A popular choice for the embedding function $\mathbf{h}_\theta$ are continuous convolutions[1,5,26] or an attention mechanism[4]. For the envelope functions Hermann et al.[1] proposed to use orbitals obtained from a Hartree-Fock calculation, whereas Pfau et al.[2] relied on exponentially decaying envelopes, (i.e., $\varphi_k(\mathbf{r}) = \exp(-\alpha_{kI}|\mathbf{r} - \mathbf{R}|)$ with trainable parameter $\alpha_{kI} \in \mathbb{R}$), to ensure the boundary conditions far away from the nuclei. In our architecture, we mainly focus on the matrix $\mathbf{F}^d = [\mathbf{F}_1^d, ..., \mathbf{F}_{n_{el}}^d] \in \mathbb{R}^{n_{el} \times D_{emb}}$. While the mapping by $\mathbf{F}^d$ works well for the wavefunction of a single compound, it is fundamentally unsuited to represent wavefunctions for multiple different compounds at once, since its dimension $n_{el} \times D_{emb}$ depends explicitly on the number of electrons. There are several potential options, how this challenge could be overcome. A naïve approach would be to generate a fixed number of $N_{orb} \geq n_{el}$ orbitals and truncate the output to the required number of orbitals $n_{el}$, which may differ across molecules. While simple to implement, this approach is however fundamentally limited to molecules with $n_{el} \leq N_{orb}$. Another approach is to use separate matrices $\mathbf{F}_\mathcal{G}^d$ for each molecule or geometry $\mathcal{G}$, as was done in[13], but also this approach can fundamentally not represent wavefunctions for molecules that are larger than the ones found in the training set. A third approach would be to not generate all orbitals in a single pass, but generate the orbitals sequentially in an auto-regressive manner, by conditioning each orbital on the previously generated orbitals. While this approach has been successful in other domains such as language processing, it suffers from inherently poor parallelization due to its sequential nature. A final approach−chosen in this work−is to replace the tensor $\mathbf{F}$ with a trainable function $\mathbf{f}_\theta^o(\mathbf{c}_{lk})$, which computes the

backflows based on some descriptor $\mathbf{c}_{Ik} \in \mathbb{R}^{N_{\text{basis}}}$ of the orbital $k$ to be generated:

$$\varphi_\theta^d(\mathbf{r}_i, \mathbf{R}_I, \mathbf{c}_{Ik}) = \exp\left(-|\mathbf{r}_i - \mathbf{R}_I| g_\theta^{e,d}(\mathbf{c}_{Ik})\right) \quad (4)$$

$$\Phi_{ik}^d = \sum_{I=1}^{N_{\text{atoms}}} \varphi_\theta^d(\mathbf{r}_i, \mathbf{R}_I, \mathbf{c}_{Ik}) \left\langle \mathbf{f}_\theta^{o,d}(\mathbf{c}_{Ik}), \mathbf{h}_i \right\rangle \quad (5)$$

Similar to $\mathbf{f}_\theta^o$, the trainable function $g_\theta^{e,d}$ also maps some descriptors to scalar values for each orbital. While there are several potential descriptors $\mathbf{c}_{Ik}$ for orbitals, one particularly natural choice is to use outputs of computationally cheap, conventional quantum chemistry methods such as Density Functional Theory or Hartree-Fock. We compute orbital features based on the expansion coefficients of a Hartree-Fock calculation, by using orbital localization and a graph convolutional network (GCN), as outlined in the methods section "Obtaining orbital descriptors from Hartree-Fock". We then map these features to orbitals $\Phi_{ik}^d$, which we call in the following *transferable atomic orbitals (TAOs)*, using odd and even functions $\mathbf{f}_\theta^o$ and $g_\theta^e$ as illustrated in Fig. 1.

## Properties of our ansatz

These TAOs fulfill many properties, which are desirable for a wavefunction ansatz:

- **Constant parameter count**: In principle, the number of parameters in the ansatz is independent of system size. While it might still be necessary to increase the parameter count to maintain uniform accuracy for systems of increasing size, TAOs have no explicit relationship between parameter count and system size. This is in contrast to previous approaches[1,2,13] where the number of parameters grows explicitly with the number of particles, making it impossible to use a single ansatz across systems of different sizes. In particular, backflows and envelope exponents have typically been chosen as trainable parameters of shape $[N_{\text{orb}} \times N_{\text{det}}]$. In our ansatz the backflows $\mathbf{F}$ are instead computed by a single function $\mathbf{f}_\theta$ from multiple inputs $\mathbf{c}_{Ik}$.

- **Equivariant to sign of HF-orbital**: Orbitals of a HF-calculation are obtained as eigenvectors of a matrix and are thus determined only up to their sign (or their phase in the case of complex eigenvectors). We enforce that the functions $\mathbf{f}_\theta^o, g_\theta^e$ are odd and even with respect to $\mathbf{c}_{Ik}$. Therefore our orbitals $\Phi_{ik}^d$ are equivariant to a flip in the sign of the HF-orbitals used as inputs: $\Phi(-\mathbf{c}_{Ik}) = -\Phi(\mathbf{c}_{Ik})$. Therefore during supervised pre-training, the undetermined sign of the reference orbitals becomes irrelevant,

leading to faster convergence as demonstrated in "Equivariance with respect to HF-phase".

- **Approximate locality**: When using localized HF-orbitals as input, the resulting TAOs are typically also localized. Localized HF-orbitals are orbitals which have non-zero orbital coefficients $\tilde{\boldsymbol{\alpha}}_{Ik}$ only on some subset of atoms. Using our architecture outlined in "Obtaining orbital descriptors from Hartree-Fock", this typically translates into local orbital features $\mathbf{c}_{Ik}$. Since we enforce the backflow $\mathbf{f}_\theta^o$ to be odd (and thus $\mathbf{f}_\theta^o(\mathbf{0}) = \mathbf{0}$), the resulting TAOs have zero contribution from atoms $I$ with $\mathbf{c}_{Ik} = \mathbf{0}$. While the true wavefunction might not be fully decomposable into purely local contributions, many relevant chemical concepts —such as core electrons, bonds, lone pairs, or chemical groups— are intrinsically local concepts and locality has been successfully used as prior in many applications, for example in Neural Network Potentials[27,28]. This hints at the prior of using local orbitals to compose many-body wavefunctions, which in turn can still contain non-local electron-electron correlations captured via the fully connected, non-local electron embedding $\mathbf{h}_i$.

- **High expressivity**: We empirically find that our ansatz is sufficiently expressive to model ground-state wavefunctions to high accuracy. We demonstrate this both empirically (c.f. SI 1, 1 mHa energy deviation against PsiFormer for NH₃) and theoretically (c.f. SI 2) in the supplementary information. This stands in contrast to previous approaches based on incorporating ab-initio orbitals[1], which could not reach chemical accuracy even for small molecules.

## Size consistency of the ansatz

One design goal of the ansatz is to allow transfer of weights from small systems to larger systems. In particular, if a large system consists of many small previously seen fragments, one would hope to obtain an energy which corresponds approximately to the sum of the fragment energies. One simple test case, are chains of equally spaced Hydrogen atoms of increasing lengths. These systems have been studied extensively using high-accuracy methods[29], because they are small systems which already show strong correlation and are thus challenging to solve. We test our method by pre-training our ansatz on chains of length 6 and 10, and then evaluating the model (with and without subsequent fine-tuning) for chain lengths between 2 and 28. Figure 2a shows that our ansatz achieves very high zero-shot-accuracy in the interpolation regime ($N_{\text{atoms}} = 8$) and for extrapolation to slightly larger or slightly smaller chains ($N_{\text{atoms}} = 4, 12$). Even when extrapolating to systems of twice the size ($N_{\text{atom}} = 20$), our method still outperforms a Hartree-Fock calculation and eventually converges to an energy close to the Hartree-Fock solution.

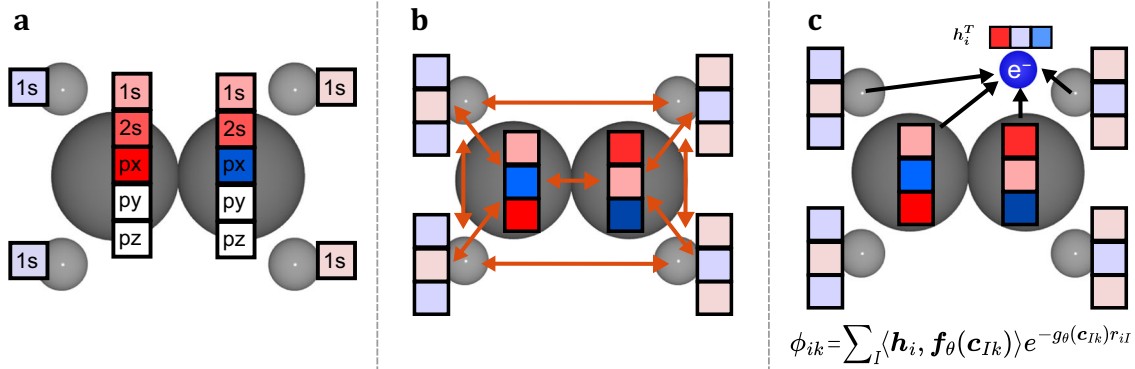

$$\phi_{ik} = \sum_I \langle \boldsymbol{h}_i, \boldsymbol{f}_\theta(\boldsymbol{c}_{Ik}) \rangle e^{-g_\theta(\boldsymbol{c}_{Ik}) r_{iI}}$$

**Fig. 1 | Illustration of the Transferable Atomic Orbitals, demonstrated on the C=C-bond of Ethene. a** The input for each orbital are localized Hartree-Fock basis expansion coefficients $\tilde{\boldsymbol{\alpha}}_I$, corresponding to every atom $I$. **b** We learn a representation $\mathbf{c}_I$ of the orbital on every atom using a Graph Neural Network, exchanging information across atoms. **c** The orbital $\phi$ is evaluated for electron $i$ by combining the electron-embedding $\mathbf{h}_i$ with the functions of the orbital representation $f_\theta(\mathbf{c}_I)$ and $g_\theta(\mathbf{c}_I)$.

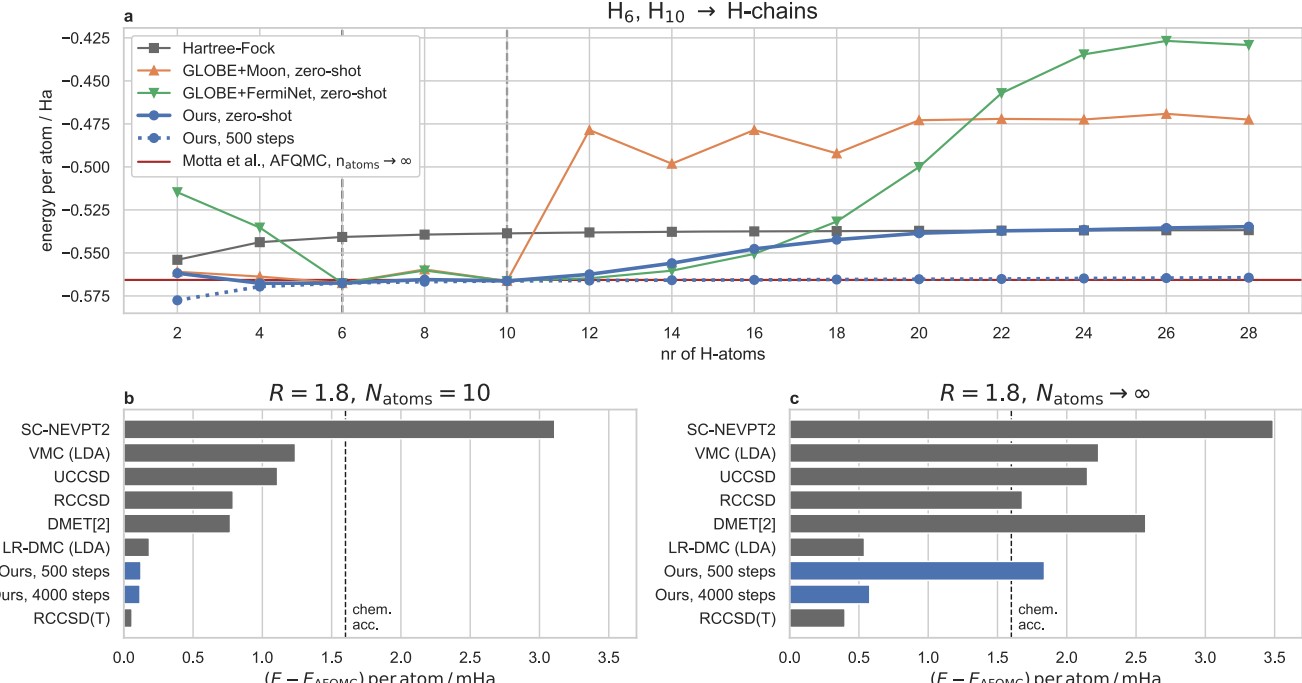

**Fig. 2 | Transferability of the ansatz to chemically similar, larger systems, demonstrated on the example of hydrogen-chains. a** Energy per atom as a function of chain-length. While GLOBE cannot successfully transfer to larger chains, our ansatz successfully predicts zero-shot energies (i.e. without fine-tuning) for up to 2x longer chains. **b**, **c** Comparison of our energies per atom after 500 and 4000 fine-tuning steps vs. high-accuracy methods from Motta et al.[29]. The methods compared in this work include: SC-NEVPT2, the strongly contracted variant of the $n_{el}$ electron valence state second-order pertubation theory; VMC (LDA), variational

Monte Carlo (local density approximation); UCCSD, couple cluster theory with full treatment of singles and doubles excitations; RCCSD and RCCSD(T), couple cluster theory with full treatment of singles and doubles and perturbative treatment of triple excitations using restricted Hartree-Fock as a reference state; DMET density-matrix embedding theory, LR-DMC (LDA) lattice-regularized diffusion Monte Carlo (local density approximation). **b** for the Hydrogen Chain with number of atoms $N_{atoms} = 10$ (**c**) for chain lengths extrapolated to the thermodynamic limit ($N_{atoms} \rightarrow \infty$).

To reach the accuracy of other correlated methods, we need a few fine-tuning steps for each new system. In Fig. 2b, c, we compare our results after 500 and 4000 fine-tuning steps against all high-accuracy methods from Motta et al.[29], which can obtain energies extrapolated to the thermodynamic limit (TDL). We compare for the two system sizes, investigated in ref. 29: 10 atoms in Fig. 2b, and extrapolation to $N_{atoms} = \infty$ in Fig. 2c. For $H_{10}$, our method is in near perfect agreement with their reference method AFQMC, deviating only by 0.1 mHa, nearly independent of the number of fine-tuning steps. This high-accuracy result is expected, since our model has also been pre-trained on chains of length 10 (albeit with different inter-atomic distances), and DL-VMC has previously been shown to achieve very high-accuracy on this system[1]. When extrapolating to the TDL, our zero-shot energies are not competitive with high-accuracy methods, but instead yield energy errors comparable to Hartree-Fock, as seen in Fig. 2a. However, fine-tuning the ansatz for only 500 steps, yields energies that already outperform most methods studied in ref. 29 and fine-tuning for 4000 steps yields a deviation of 0.6 mH/atom vs. AFQMC, on par with specialized methods such as LR-DMC.

This good performance stands in stark contrast to other approaches such as GLOBE+ FermiNet or GLOBE+Moon, studied in ref. 26: Both GLOBE-variants yield 5-6x higher errors in the interpolation regime and both converge to much higher energies for larger chains. While our approach yields Hartree-Fock-like energies for very long chains, GLOBE+FermiNet and GLOBE+Moon yield results that are outperformed even by assuming a chain of non-interacting H-atoms, which would yield an energy per atom of -0.5 Ha. For modest extrapolations ($N_{atoms} = 12$ to $N_{atoms} = 20$) our zero-shot results yield 3–20x lower errors than GLOBE+Moon.

### Equivariance with respect to HF-phase

Due to using even and odd functions for the TAOs, our orbitals are equivariant with respect to a change of sign of the Hartree Fock orbitals. Therefore, a sign change of the HF-orbitals during HF-pre-training has no effect on the optimization of the wavefunction. One test case to assure this behavior is the rotation of a $H_2O$ molecule, where we consider a set of 20 rotations of the same geometry, leading to a change of sign in the p-orbitals of the Oxygen atom (cf. Fig. 3). We evaluate our proposed architecture and compare it against a naïve approach, where we use a standard backflow matrix **F**, instead of a trainable, odd function $\mathbf{f}_\theta^o$. In Fig. 3 we can see a clear spike in the HF-pre-training loss at the position of the sign flip for the standard backflow-type architecture, causing slower convergence during the subsequent variational optimization. After 16k optimization steps the effect diminishes and no substantial improvement on the accuracy can be observed. Although in this specific instance the orbital sign problem could also be overcome without our approach by correcting the phase of each orbital to align them across geometries, phase alignment is not possible in all circumstances. For example, there are geometry trajectories, where the Berry phase prevents such solutions[30].

### Transfer to larger, chemically similar compounds

To test the generalization and transferability of our approach, we perform the following experiment: First, we train our ansatz on a dataset of multiple geometries of a single, small compound (e.g. 20 distorted geometries of Methane). For this training, we follow the usual procedure of supervised HF-pre-training and subsequent variational optimization as outlined in the methods section "Variational Monte Carlo". After 64k variational optimization steps, we

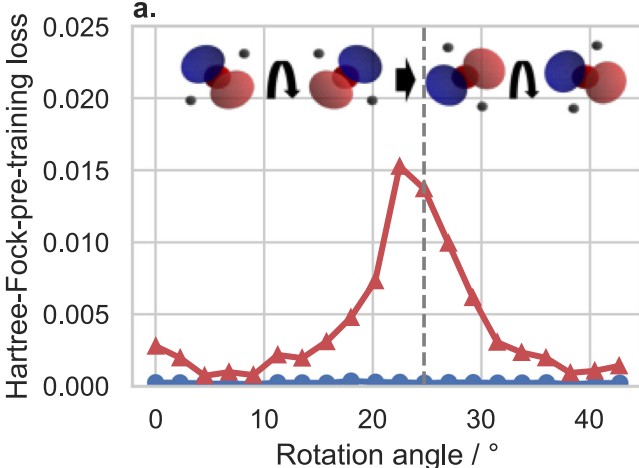

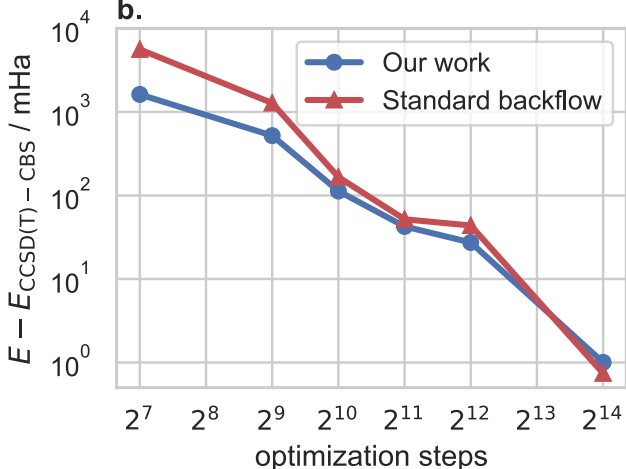

**Fig. 3 | Accuracy when Hartree-Fock-pre-training against rotated $H_2O$ molecules, which contain a change of sign in the Hartree-Fock-p-orbitals of the Oxygen atom (a).** Comparing a shared optimization of a backflow-based neural network wavefunction (Standard backflow) against transferable atomic orbitals (Our work). **a** Hartree-Fock-pre-training loss of the last the last 100 Monte Carlo samples for 20 rotated geometries (**b**): Mean energy error vs. couple cluster reference calculations ($E_{CCSD(T)}$), averaged across all geometries.

then re-use the weights for different geometries of a larger compound (e.g. distorted geometries of Ethene). We fine-tune the model on this new geometry dataset for a variable number of steps and plot the resulting energy errors in Fig. 4. We do not require supervised HF-pre-training on the new, larger dataset. We perform this experiment for 3 pairs of test systems: Transferring from geometries of Hydrogen-chains with 6 atoms each, to chains with 10 atoms each, transferring from Methane to Ethene, and transferring from Ethene to Cyclobutadiene. These test systems are of interst, because they show strong correlation, despite being relatively small and computationally cheap systems. For example, even CCSD(T) overestimates the energy barriers of the Ethene- and Cyclobutadiene-PES by ≈10 mHa[13,31].

We compare our results to the earlier DeepErwin approach[13], which only partially reused weights, and GLOBE, a concurrent work[26] which reuses all weights. To measure accuracy we compare two important metrics: First, the mean energy error (averaged across all geometries $g$ of the test dataset) $\frac{1}{N}\sum_g(E_g - E_g^{ref})$, which reflects the method's accuracy for absolute energies (cf. Fig. 4a). Second, the deviation of the relative energy between the highest and lowest point of the PES, i.e. $\Delta E - \Delta E_{ref} = (E_{max} - E_{min}) - (E_{max}^{ref} - E_{min}^{ref})$, plotted in Fig. 4b. Since different studies use different batch-sizes and different

definitions of an epoch, we plot all results against the number of samples used for the energy estimation during variational optimization, which is very closely linked to computational cost.

Compared to other approaches, we find that our method yields substantially lower and more consistent energies. On the toy problem of $H_6$ to $H_{10}$ our approach and GLOBE reach the same accuracy, while DeepErwin converges to higher energies. For the real-world molecules Ethene ($C_2H_4$) and Cyclobutadiene ($C_4H_4$) our approach reaches substantially lower (and thus more accurate) energies and much more consistent potential energy surfaces. After 64 mio. fine-tuning samples, our mean absolute energies are 16 mHa and 17 mHa lower than GLOBE, and our relative energies are 39 mHa and 20 mHa closer to the reference calculation. When inspecting the resulting Potential Energy Surface for Ethene (Fig. 4c), we find that we obtain qualitatively similar results as DeepErwin and MRCI, but obtain energies that are ≈ 6 mHa lower (and thus more accurate). GLOBE on the other hand does not yield the correct PES for this electronically challenging problem, since it overestimates the energy barrier at ≈90° twist angle by ≈50 mHa. We observe similar results on the Cyclobutadiene geometries, where our approach yields relative energies that are in close agreement to the reference method, while the GLOBE-results overestimate the energy difference by ≈20 mHa.

### Towards a first foundation model for neural network wavefunctions

While the experiments in the previous section demonstrate the ability to pre-train our model and fine-tune it on a new system, the resulting pre-trained models are of little practical use, since they are only pre-trained on a single compound each and can thus not be expected to generalize to chemically different systems. To obtain a more diverse pre-training dataset, we compiled a dataset of 360 distorted geometries, spread across 18 different compounds. The dataset effectively enumerates all chemically plausible molecules with up to 18 electrons containing the elements H, C, N, and O. For details on the data generation see ""Dataset used for pre-training of multi-compound model". We pre-train a base-model for 500,000 steps on this diverse dataset and subsequently evaluate its performance, when computing Potential Energy Surfaces. We evaluate its error against CCSD(T) (extrapolated to the complete basis set limit) both for compounds that were in the pre-training dataset (with different geometries), as well as for new, larger, out-of-distribution compounds which were not present in the pre-training dataset. We compare the results against a baseline model, which uses the same architecture, but is trained from scratch. Instead of re-using the pre-trained weights, this baseline initializes its weights using the default method of supervised HF-pre-training for each specific molecule[2].

Figure 5 shows that fine-tuning our pre-trained model yields substantially lower energies than the usual optimization from a HF-pre-trained model. For example, for new large compounds, it only takes 1k fine-tuning steps of the pre-trained model, to reach the same accuracy as CCSD(T) with a 3Z basis set. The non-pretrained model has a 60x higher energy error after 1k optimization steps, and requires 20x more steps to reach this accuracy. As expected, the gains from pre-training diminish for long subsequent optimization, but after 32k optimization steps, the pre-trained model still demonstrates 3x lower energy errors than the model being trained from scratch.

To assess the accuracy of our method for relative energies, we use the pre-trained model to compute a potential energy surface of a carbon dimer. Figure 6 compares our energies (with and without fine-tuning) against other state-of-the art conventional and deep-learning-based methods. We compare against CCSD(T) extrapolated to the complete basis-set limit, an FCI-QMC study of

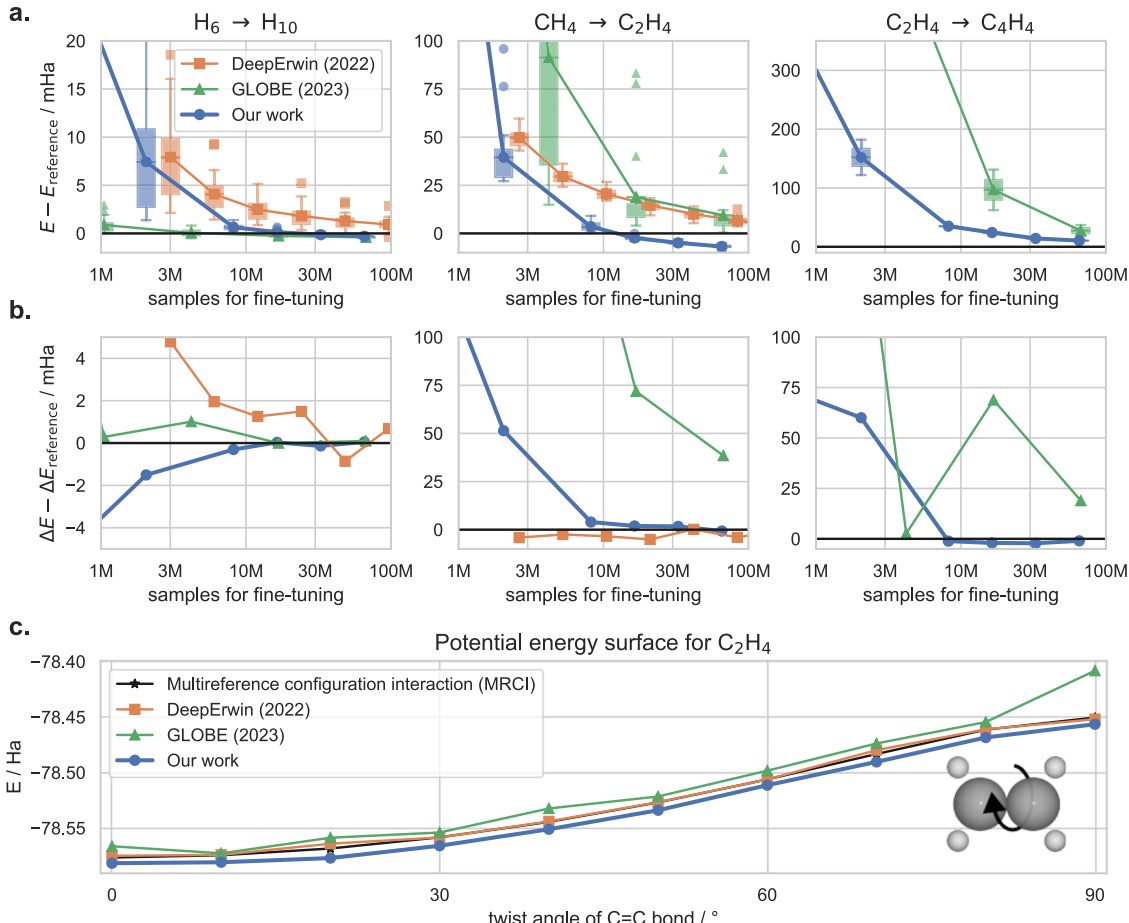

**Fig. 4 | Accuracy when pre-training the model on small compounds and reusing it for larger compounds.** Boxplots show the 25-75th percentile of energy deviations, connecting lines show mean energy deviations, whiskers span the non-outlier range (1.5 interquartile ranges above and below the boxes), energy deviations beyond the whiskers are plotted individually. **a** Mean energy vs reference energy $E_{ref}$, averaged across all geometries of the test set. **b** Deviation of relative energies. **c** Final Potential Energy Surface (PES) for the Ethene molecule for each method. For the H-chains and $C_2H_4$ we use MRCI results from[13] as reference energy, for Cyclobutadiene we use FermiNet results from[16] as reference.

the $C_2$-dimer by Booth et al.[32] and PsiFormer[4], the currently most accurate deep-learning based ansatz for absolute energies. We find that our approach without any fine-tuning steps correctly identifies the energy minimum at $d = 2.35$ bohr and even yields equilibrium energies that are lower than FCIQMC (cf. Fig. 6a). While the carbon dimer itself is not part of the pre-training dataset, several molecules with $C = C$ bonds are, which explains the relatively high accuracy in this regime. When stretching the bond, our zero-shot energies overestimate the resulting energy by roughly 250mHa (cf. Fig. 6b), clearly highlighting this failure case in a regime of lacking pre-training data. However, after just 1k fine-tuning steps of our pre-trained base-model, we obtain the qualitatively correct PES. In particular in the electronically most challenging regime around $d = 3$ bohr, FCIQMC and CCSD(T) both systematically overestimate the relative energy by 20-30 mHa compared to PsiFormer, wheres our method only overestimates the energy by ca. 10 mHa (cf. Fig. 6c). We note that our approach uses only 1k optimization steps per geometry, compared to 100k (and twice the batch size) for PsiFormer, thus requiring GPU-hours instead of GPU-days.

## Scaling behavior
In many domains, increasing the amount of pre-training, has led to substantially better results, even without qualitative changes to the architecture[33]. To investigate the scalability of our approach, we vary the three key choices, along which one could increase the scale of pre-

training: The size of the wavefunction model, the number of compounds and geometries present in the pre-training-dataset, and the number of pre-training steps. Starting from a large model, trained on 18x20 geometries, for 256k pre-training steps, we independently vary each parameter. We test 3 different architectures sizes, with decreasing layer width and depth for the networks $f_\theta$, $g_\theta$, and $GCN_\theta$ (Computational settings). We test 3 different training sets, with decreasing number of compounds in the training set, with 20 geometries each (Dataset used for pre-training of foundationmodel). Finally, we evaluate model-checkpoints at different amounts of pre-training, ranging from 64k steps to 512k steps. Figure 7 depicts the accuracy obtained by subsequently fine-tuning the resulting model for just 4000 steps on the evaluation set. In each case, increasing the scale of pre-training clearly improves evaluation results—both for the small in-distribution compounds, as well as the larger out-of-distribution compounds. We find a strong dependence of the accuracy on the model size and number of compounds in the pre-training dataset, and a weaker dependency on the number of pre-training steps. While our computational resources, currently prohibit us from training at larger scale, the results indicate that our approach may already be sufficient to train an accurate multi-compound, multi-geometry foundation model for wavefunctions.

## Discussion
This work presents an ansatz for deep-learning-based VMC, which can in principle be applied to molecules of arbitrary size. We

demonstrate the favorable properties of our ansatz, such as extensivity, zero-shot prediction of wavefunctions for similar molecules (Size consistency of the ansatz), invariance to the phase of orbitals (Equivariance with respect to HF-phase) and fast fine-tuning for larger, new molecules (Transfer to larger, chemically similar compounds). Most importantly, "Towards a first foundation model for neuralnetwork wavefunctions" is, to our knowledge, the first successful demonstration of a wavefunction, which is transferable across compounds and has successfully been trained on a diverse dataset of compounds and geometries. We demonstrate that the dominating deep-learning paradigm of the last years—pre-

training on large data and fine-tuning on specific problems—can also be applied to the difficult problem of wavefunctions. While previous attempts[13,26] have failed to obtain high-accuracy energies from pre-trained neural network wavefunctions, we find that our approach yields accurate energies and does so at a fraction of the cost needed without pre-training. A typical inference run (batch-size 2048, one compute node with 2 GPUs) for a molecule with 3 heavy atoms takes ~30 min for zero-shot evaluation, or 1.5h for 1k fine-tuning steps and subsequent evaluation. A CCSD(T)-4Z calculation on a compute-node with 128 CPUs took ~30 min. Given that the per-iteration cost of DL-VMC scales as $\mathcal{O}(n_{el}^4)$ vs. the $\mathcal{O}(n_{el}^7)$ cost of CCSD(T), we expect our model to become competitive once pre-trained and applied to sufficiently large molecules. We furthermore demonstrate in "Scaling behavior" that results can be improved systematically by scaling up any aspect of the pre-training: Model size, data-size, or pre-training-steps.

Despite these promising results, there are many open questions and limitations which should be addressed in future work. First, we find that our ansatz currently does not fully match the accuracy of state-of-the-art single-geometry DL-VMC ansätze. While our approach consistently outperforms conventional variational methods such as MRCI or CCSD at finite basis set, larger, computationally more expensive DL-VMC models can reach even lower energies. For example, PsiFormer optimized for 100k steps on the carbon dimer, reaches ≈20 mHa lower absolute energies than our approach fine-tuned for 1k steps. Exchanging our message-passing-based electron-embedding, with recent attention based approaches[4] should lead to higher accuracy. Furthermore we have made several deliberate design choices, which each trade-off expressivity (and thus potentially accuracy) for computational cost: We do not exchange information across orbitals and we base our orbitals on computationally cheap HF-calculations. Including attention or message passing across orbitals (e.g. similar to ref. 26), and substituting HF for a trainable, deep-learning-based model should further increase expressivity. While we currently use HF-orbitals due to their widespread use and low computational cost, our method does not rely on a specific orbital descriptor. We could substitute HF for a separate model such as PhisNet[20] or SchnOrb[34] to compute orbital descriptors $c_{lk}$, leading to a fully end-to-end machine-learned wavefunction. Second, while we include useful physical priors such as locality, we do not yet currently use the invariance of the Hamiltonian with respect to rotations, inversions or spin-flip. E3-equivariant networks have been highly successful for neural network force-fields, but have not yet been applied to

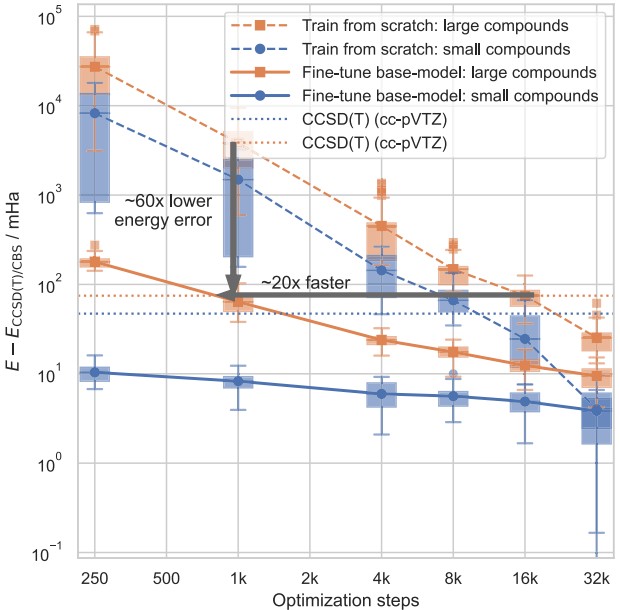

**Fig. 5 | Fine-tuning of variationally pre-trained base-model (solid lines) vs. training a model from scratch (dashed lines) for 70 different geometries.** Boxplots show the 25-75th percentile of energy deviations, connecting lines show mean energy deviations, whiskers span the non-outlier range (1.5 interquartile ranges above and below the boxes), energy deviations beyond the whiskers are plotted individually. Small compounds are in-distribution, with geometries similar to geometries in pre-training dataset. Larger compounds are out-of-distribution and are not present in the pre-training dataset.

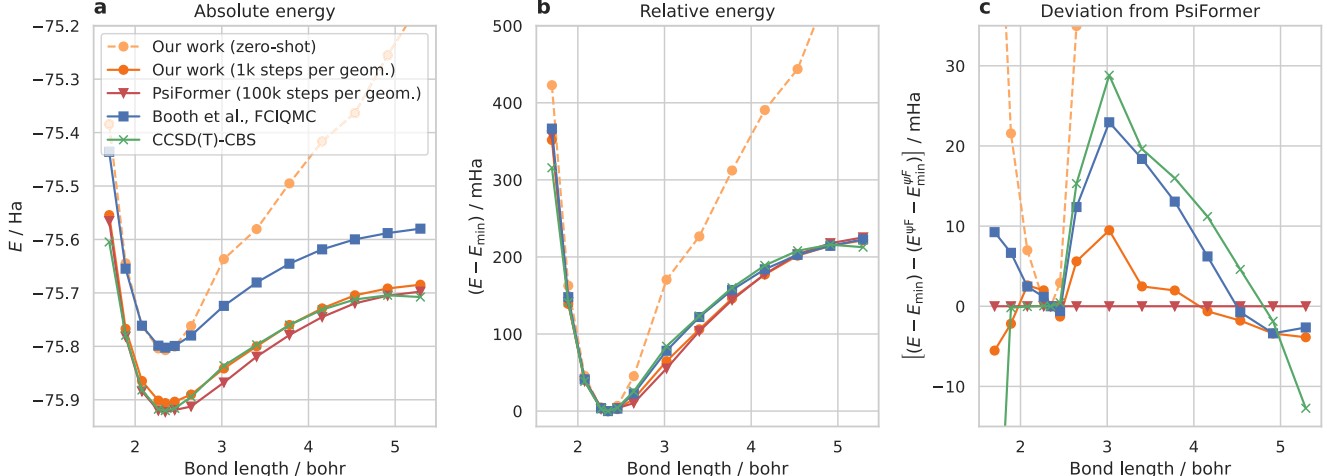

**Fig. 6 | Potential energy surface of $C_2$. a** Absolute energies, (**b**) Energies of each method relative to the energy minimum at $d = 2.35$ bohr, (**c**) Deviation of relative energies from the relative energies obtained by PsiFormer ($\psi F$).

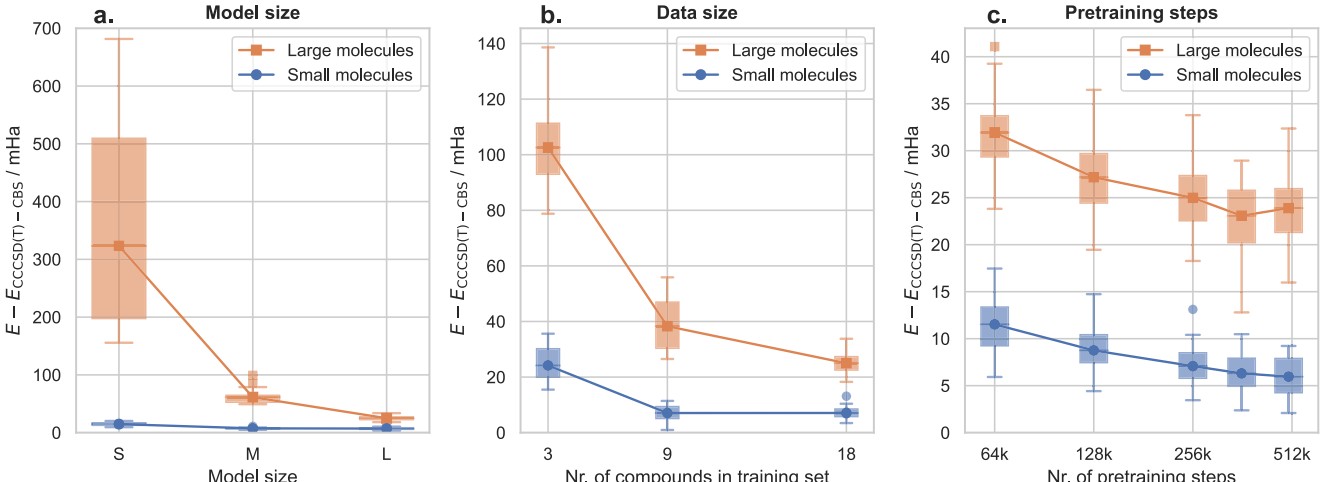

**Fig. 7 | Error when fine-tuning the pre-trained model for 4000 steps on small in-distribution geometries and larger out-of-distribution geometries.** Boxplots show the 25-75th percentile of energy deviations, connecting lines show mean energy deviations, whiskers span the non-outlier range (1.5 interquartile ranges above and below the boxes), energy deviations beyond the whiskers are plotted individually. **a** The energy error for increasing the model size of the transferable atomic orbitals. The small model uses no hidden layers and no graph convolutional network. The medium-sized model uses one hidden layer of width 64 and 128 for $g_\theta$ and for $f_\theta$ respectively, and one iteration of the graph convolutional network. The large model uses two iterations of the graph convolutional network, $f_\theta$ and $g_\theta$ with hidden dimension 256 and 128. **b** The energy error when using a pre-trained model on a dataset with either 3, 9 or 18 compounds. **c** The energy error with increasing number of pretraining steps.

wavefunctions due to the hitherto unsolved problem of symmetry breaking[15]. Using HF-orbitals as symmetry breakers, could open a direct avenue towards E3-equivariant neural network wavefunctions. Third, while we use locality of our orbitals as a useful prior, we do not yet use it to reduce computational cost. By enforcing sparsity of the localized HF-coefficients, one could limit the evaluation of orbitals to a few participating atoms, instead of all atoms in the molecule. While the concurrent GLOBE approach enforces its orbitals to be localized at a single position, our approach naturally lends itself to force localization at a given number of atoms, allowing for a deliberate trade-off of accuracy vs. computational cost. Lastly, we observe that our method performs substantially better, when dedicating more computational resources to the pre-training, which makes it likely that future work will be able to scale up our approach. To facilitate this effort we open source our code, dataset as well as model parameters.

## Methods
### Variational Monte Carlo
Considering the Born-Oppenheimer approximation, a molecule with $n_{el}$ electrons and $N_{atoms}$ nuclei can be described by the time-independent Schrödinger equation

$$\hat{H}\psi = E\psi \tag{6}$$

with the Hamiltonian

$$\hat{H} = -\frac{1}{2}\sum_i \nabla^2_{\mathbf{r}_i} + \sum_{i>j}\frac{1}{|\mathbf{r}_i - \mathbf{r}_j|} + \sum_{I>J}\frac{Z_I Z_J}{|\mathbf{R}_I - \mathbf{R}_J|} - \sum_{i,I}\frac{Z_I}{|\mathbf{r}_i - \mathbf{R}_I|} \tag{7}$$

By $\mathbf{r} = (\mathbf{r}_1, \ldots, \mathbf{r}_{n_\uparrow}, \ldots, \mathbf{r}_{n_{el}}) \in \mathbb{R}^{3 \times n_{el}}$ we denote the set of electron positions divided into $n_\uparrow$ spin-up and $n_\downarrow = n_{el} - n_\uparrow$ spin-down electrons. The solution to the electronic Schrödinger equation $\psi$ needs to fulfill the anti-symmetry property, i.e. $\psi(\mathcal{P}\mathbf{r}) = -\psi(\mathbf{r})$ for

any permutation $\mathcal{P}$ of two electrons of the same spin. Finding the groundstate wavefunction of a system corresponds to finding the solution to Eq. (6), with the lowest eigenvalue $E_0$. Using the Rayleigh-Ritz principle, an approximate solution can be found through minimization of the loss

$$\mathcal{L}(\psi_\theta) = \mathbb{E}_{\mathbf{r} \sim \psi_\theta^2(\mathbf{r})}\left[\frac{(\hat{H}\psi_\theta)(\mathbf{r})}{\psi_\theta(\mathbf{r})}\right] \geq E_0, \tag{8}$$

using a parameterized trial wavefunction $\psi_\theta$. The expectation value in Eq. (8) is computed by drawing samples $\mathbf{r}$ from the unnormalized probability distribution $\psi_\theta^2(\mathbf{r})$ using Markov Chain Monte Carlo (MCMC). The application of the Hamiltonian to the wavefunction can be computed using automatic differentiation and the loss is minimized using gradient based minimization. A full calculation typically consists of three steps:

(i) **Supervised HF-pre-training**: Minimization of the difference between the neural network ansatz and a reference wavefunction (e.g. a Hartree-Fock calculation) $||\psi_\theta - \psi^{HF}||$. This is the only part of the procedure which requires reference data, and ensures that the initial wavefunction roughly resembles the true groundstate. While this step is in principle not required, it substantially improves the stability of the subsequent variational optimization.

(ii) **Variational optimization**: Minimization of the energy (Eq. (8)) by drawing samples from the wavefunction using MCMC, and optimizing the parameters $\theta$ of the ansatz using gradient based optimization.

(iii) **Evaluation**: Evaluation of the energy by evaluating Eq. (8) without updating the parameters $\theta$, to obtain unbiased estimates of the energy.

To obtain a single wavefunction for a dataset of multiple geometries or compounds, only minimal changes are required. During supervised and variational optimization, for each gradient step we pick one geometry from the dataset. We pick geometries either in a

round-robin fashion, or based on the last computed energy variance for that geometry. We run the Metropolis Hastings algorithm[35] for that geometry to draw electron positions **r** and then evaluate energies and gradients. For each geometry we keep a distinct set of electron samples **r**.

## Obtaining orbital descriptors from Hartree-Fock

As discussed in "A multi-compound wavefunction ansatz", our ansatz effectively maps uncorrelated, low-accuracy Hartree-Fock orbitals, to correlated, high-accuracy neural network orbitals. The first step in this approach is to obtain orbital descriptors $\mathbf{c}_k$ for each orbital $k$, based on a Hartree-Fock calculation.

The Hartree-Fock method uses a single determinant as ansatz, composed of single-particle orbitals $\phi_k^{HF}$:

$$\psi^{HF}(\mathbf{r}_1, \dots, \mathbf{r}_{n_{el}}) = \det\left[\Phi_{ik}^{HF}\right]_{i,k=1\dots n_{el}} \quad (9)$$

$$\Phi_{ik}^{HF} := \phi_k^{HF}(\mathbf{r}_i) \quad (10)$$

For molecules, these orbitals are typically expanded in atom-centered basis-functions $\mu(\mathbf{r})$, with $N_{basis}$ functions centered on each atom $I$:

$$\phi_k^{HF}(\mathbf{r}) = \sum_{I=1}^{N_{atoms}} \sum_{b=1}^{N_{basis}} \alpha_{Ik,b}\, \mu_b(\mathbf{r} - \mathbf{R}_I), \quad (11)$$

The coefficients $\boldsymbol{\alpha}_{Ik} \in \mathbb{R}^{N_{basis}}$ and the corresponding orbitals $\phi_k^{HF}(\mathbf{r})$ are obtained as solutions of an eigenvalue problem and are typically delocalized, i.e. they have non-zero contributions from many atoms. However, since $\det[U\Phi] = \det[U]\det[\Phi]$, the wavefunction is invariant under linear combination of orbitals by a matrix $U$ with $\det[U] = 1$. One can thus choose localized orbital expansion coefficients

$$\tilde{\alpha}_{Ik,b} = \sum_{k'=1}^{N_{orb}} \alpha_{Ik,b} U_{kk'} \quad (12)$$

corresponding to orbitals which are maximally localized according to some metric. We stress that such a transformation from canonical to localized orbitals is lossless: The localized orbitals represent exactly the same wavefunction as the canonical orbitals and thus the procedure involves no approximation. We localize orbitals purely to simplify the learning problem for the subsequent trainable functions **f**, and $g$, which map orbital descriptors to backflows and exponents. If the orbitals for typical molecules can be composed of recurring local motives (which empirically holds true), this substantially simplifies the generalization of **f** and $g$ to larger molecules, since their inputs will mostly consist of orbital coefficients already seen in smaller molecules. Several different metrics and corresponding localization schemes, such as Foster-Boys[36] or Pipek-Mezey[37], have been proposed to find the optimal transformation matrix $U$ and are easily available as computationally cheap post-processing options in quantum chemistry codes. We use the Foster-Boys method as implemented in pySCF[38].

Due to the fundamentally local nature of atom-wise orbital coefficients $\tilde{\boldsymbol{\alpha}}_{Ik}$, which can be insufficient to distinguish orbitals, we use a fully connected graph convolutional neural network (GCN) to add context about the surrounding atoms. We interpret each atom as a node (with node features $\tilde{\boldsymbol{\alpha}}_{Ik}$) and use the set of all 3D inter-atomic distance vectors $\{\mathbf{R}_{IJ}\}$ as edge features:

$$\mathbf{c}_{Ik} = \mathbf{GCN}_{\theta,I}\left(\{\tilde{\boldsymbol{\alpha}}_{Jk}\}_{J=1\dots N_{atoms}}, \{\mathbf{R}_{JJ'}\}\right),$$
$$J J' = 1\dots N_{atoms} \quad$$

We embed the edge features using a Kronecker product of Gaussian basis functions (of means $\boldsymbol{\mu} \in \mathbb{R}^{D_{edge}}$ and widths $\boldsymbol{\sigma} \in \mathbb{R}^{D_{edge}}$) of the inter-atomic distance $R_{IJ}$ and the concatenation of the 3D-distance vector with the constant 1. The embedded edge features are then mapped to a high-dimensional feature space with a multi-layer perceptron (MLP):

$$\tilde{\mathbf{e}}_{IJ} = \exp\left(-\frac{(\mathbf{R}_{IJ} - \boldsymbol{\mu})^2}{2\boldsymbol{\sigma}^2}\right) \otimes [1|\mathbf{R}_{IJ}] \quad (13)$$

$$\mathbf{e}_{IJ} = \text{MLP}(\tilde{\mathbf{e}}_{IJ}) \quad (14)$$

$$\begin{aligned} \mathbf{c}_{Ik}^0 &= \tilde{\boldsymbol{\alpha}}_{Ik} \\ \tilde{\mathbf{e}}_{IJ} &\in \mathbb{R}^{4D_{edge}}, \quad \mathbf{c}_{Ik}^0 \in \mathbb{R}^{N_{basis}} \end{aligned} \quad (15)$$

Each layer $l$ of the GCN consist of the following update rules

$$\mathbf{u}_{Ik}^l = \sum_J \mathbf{c}_{Jk}^l \odot \left(\mathbf{W}_e{}^l \mathbf{e}_{IJ}\right), \quad (16)$$

$$\mathbf{c}_{Ik}^{l+1} = \sigma\left(\mathbf{W}_c{}^l \mathbf{c}_{Ik}^l + \mathbf{W}_u{}^l \mathbf{u}_{Ik}^l\right), \quad (17)$$

with trainable weight matrices $\mathbf{W}_e{}^l$, $\mathbf{W}_c{}^l$, $\mathbf{W}_u{}^l$ and the SiLU activation function $\sigma$[39]. After $L$ iterations we use the final outputs as orbitals features:

$$\mathbf{c}_{Ik} := \mathbf{c}_{Ik}^L \quad (18)$$

## Mapping orbital descriptors to wavefunctions

To obtain entries $\Phi_{ik}$ of the Slater determinant, we combine a high-dimensional electron embedding $\mathbf{h}_i$ with a function of the orbital descriptor $\mathbf{c}_{Ik}$:

$$\mathbf{h}_i = \mathbf{h}_\theta(\mathbf{r}_i, \{\mathbf{r}_\uparrow\}, \{\mathbf{r}_\downarrow\}, \{(\mathbf{R}, \mathbf{Z})\}) \quad (19)$$

$$\varphi_\theta^d(\mathbf{r}_i, \mathbf{R}_I, \mathbf{c}_{Ik}) = \exp\left(-|\mathbf{r}_i - \mathbf{R}_I|\, g_\theta^{e,d}(\mathbf{c}_{Ik})\right) \quad (20)$$

$$\Phi_{ik}^d = \sum_{I=1}^{N_{atoms}} \varphi_\theta^d(\mathbf{r}_i, \mathbf{R}_I, \mathbf{c}_{Ik}) \left\langle \mathbf{f}_\theta^{o,d}(\mathbf{c}_{Ik}), \mathbf{h}_i \right\rangle \quad (21)$$

The functions $\mathbf{GCN}_\theta^o$, $\mathbf{f}_\theta^o$, and $g_\theta^e$ are trainable functions, which are enforced to be odd and even with respect to change in sign of their argument **c**:

$$\begin{aligned} &\text{Even } g_\theta^e: \\ &g_\theta^e(\mathbf{c}) := g_\theta(\mathbf{c}) + g_\theta(-\mathbf{c}) \end{aligned} \quad (22)$$

$$\begin{aligned} &\text{Odd } \mathbf{f}_\theta^o: \\ &\mathbf{f}_\theta^o(\mathbf{c}) := \mathbf{f}_\theta(\mathbf{c}) - \mathbf{f}_\theta(-\mathbf{c}) \end{aligned} \quad (23)$$

$$\begin{aligned} &\text{Odd } \mathbf{GCN}_\theta^o: \\ &\mathbf{GCN}_\theta^o(\boldsymbol{\alpha}, \mathbf{R}) := \mathbf{GCN}_\theta(\boldsymbol{\alpha}, \mathbf{R}) - \mathbf{GCN}_\theta(-\boldsymbol{\alpha}, \mathbf{R}) \end{aligned} \quad (24)$$

To obtain electron embeddings $\mathbf{h}_i$ we use the message-passing architecture outlined in[5], which is invariant with respect to permutation of electrons of the same spin, or the permutation of ions. Note that during training, all samples in a batch come from the same geometry, and thus have the same values for **R**, **Z**, and $\tilde{\boldsymbol{\alpha}}$. While the embedding

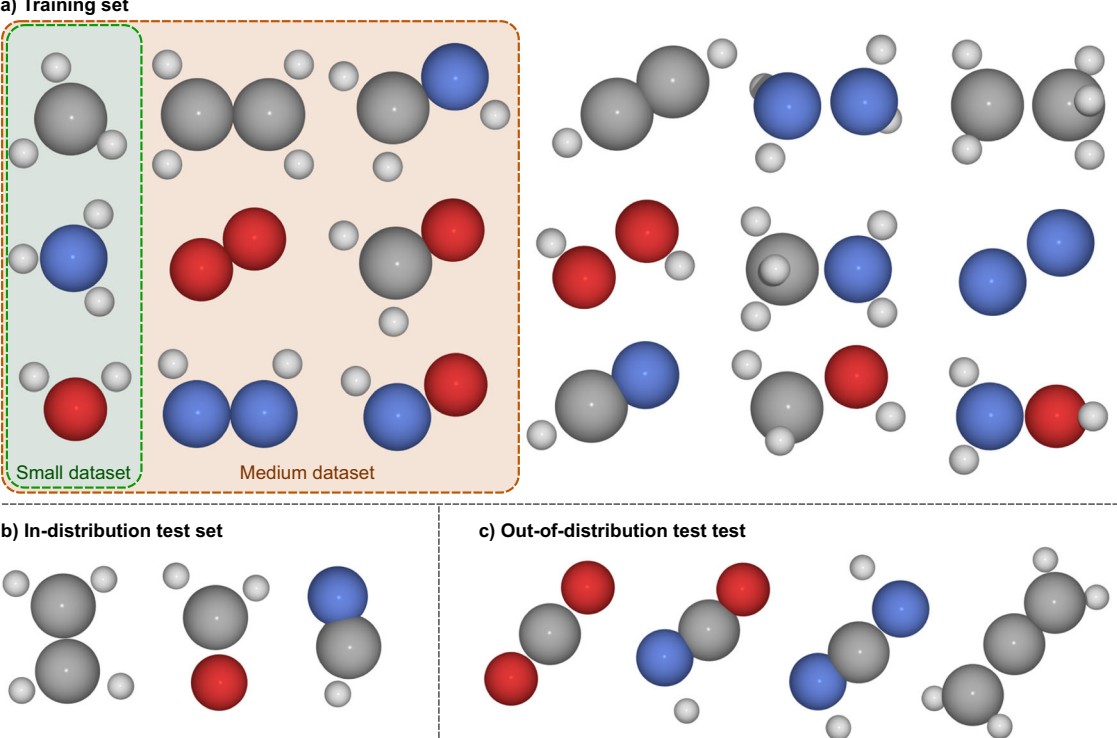

**Fig. 8 | Compounds used for pre-training and evaluation of our model.** Atom colors follow the usual convention of $H$ = white, $C$ = gray, $N$ = blue, $O$ = red. **a** The full training set, containing 18 compounds, each with 20 randomly distorted geometries. The small and medium sized training sets are subsets of this full training set, containing 3 and 9 compounds respectively. **b** The in-distribution test set consists of 3 compounds with 10 distorted geometries each. **c** The out-of-distribution test set consists of 4 compounds with 10 distortions each.

network $\mathbf{h}_\theta^{\mathrm{embed}}$, needs to be re-evaluated for every sample, the networks $\mathbf{GCN}_\theta$, $\mathbf{f}_\theta$, and $g_\theta$ only need to be evaluated once per batch, substantially reducing their impact on computational cost.

## Dataset used for pre-training of multi-compound model

We use RDKit[40] to generate all valid SMILES of molecules containing 1-3 atoms of the elements C, N, O. For each bond between atoms we allow single, double, and triple bonds. After saturating the molecules with Hydrogen, we perform force-field based geometry relaxation using RDKit. We obtain 18 compounds with 10-18 electrons, which we use for pre-training (cf. Fig. 8) and 35 compounds with 20-24 electrons, of which we use some for evaluation. Contrary to other datasets of small molecules such as GDB-7, our dataset also includes compounds which do not contain Carbon, such as the nitrogen dimer $N_2$ or hydrogen peroxide $H_2O_2$. To obtain a more diverse dataset we perturb each equilibrium geometry by applying Gaussian noise to the 3D coordinates. Since this can generate nonphysical geometries, we keep only geometries in which the perturbed inter-atomic distances are between 90–140% of the unperturbed distances.

## Reference energies

Reference energies for $H_2O$ in Fig. 3 were computed using DL-VMC for 100,000 steps[5]. Reference energies for $H_{10}$ and $C_2H_4$ in Fig. 4 were computed using MRCI-F12(Q)[13]. Reference energies for $C_4H_4$ in Fig. 4 were computed using DL-VMC[16]. To compute reference energies for our multi-compound dataset used in Fig. 5 and Fig. 7, we used pySCF[38] to perform CCSD(T) calculations using the cc-pCVXZ basis sets. We computed Hartree-Fock energies $E_X^{\mathrm{HF}}$ using basis-sets of valence $X = \{2, 3, 4\}$ and CCSD(T) energies $E_X^{\mathrm{CCSD(T)}}$ using valence $X = \{2, 3\}$. To extrapolate to the complete-basis-set-

limit, we followed[2] and fitted the following functions with free parameters $E_{\mathrm{CBS}}^{\mathrm{HF}}$, $E_{\mathrm{CBS}}^{\mathrm{corr}}$, $a, b, c$:

$$E_X^{\mathrm{HF}} = E_{\mathrm{CBS}}^{\mathrm{HF}} + ae^{-bX}$$
$$E_X^{\mathrm{corr}} := E_X^{\mathrm{HF}} - E_X^{\mathrm{CCSD(T)}} = E_{\mathrm{CBS}}^{\mathrm{corr}} + cX^{-3}$$
$$E_{\mathrm{CBS}}^{\mathrm{CCSD(T)}} = E_{\mathrm{CBS}}^{\mathrm{HF}} + E_{\mathrm{CBS}}^{\mathrm{corr}}$$

We note that neither CCSD itself, nor the perturbative (T) treatment, nor the CBS extrapolation are variational methods. The computed reference energies are therefore not variational and may underestimate the true groundstate energy.

## Computational settings

For a more detailed summary and explanation of the high-dimensional embedding structure we refer to the original work[5]. In all experiments we relied on the second order optimizer K-FAC[41,42]. Key hyperparameters used in this work are summarized in Table 1. For the base model in "Towards a first foundation model for neural network wavefunctions" we increased the initial damping by 10x and ramped it down to $1 \times 10^{-3}$ with an inverse scheduler. All runs reusing pre-trained weights, offset the learning rate scheduler by $o = 32,000$ steps, i.e. $\mathrm{lr}(t) = \mathrm{lr}_0(1+(t+o)/6000)^{-1}$. This leads to a 5x lower initial learning rate. All pre-training runs in "Transfer to larger, chemically similar compounds" used 64,000 optimization steps. The base model in "Towards a first foundation model for neural network wavefunctions" used 512,000 optimization steps due to the larger and more diverse training corpus.

The small- and medium-sized model for our ablation study in Fig. 7 differ from the large model by the number of hidden layers for $\mathbf{f}_\theta$ and $g_\theta$, the number of neurons per layer, and the number of iterations of the $\mathbf{GCN}_\theta$: The small model uses no hidden layers and

**Table 1 | Hyperparameter settings used in this work**

| | | |
|---|---|---|
| HF-pre-training | Pre-training basis set | 6-31G + p-functions for H |
| | Pre-training steps per geometry | 100-500 |
| **Embedding** | Hidden dimension of $\mathbf{h}_i$ | 256 |
| | Dimension of SchNet convolution | 32 |
| | Nº iterations embedding | 4 |
| | Activation function | tanh |
| **Transferable atomic orbitals** | Nº determinants $N_{det}$ | 4 |
| | Basis set | 6-31G + p-functions for H |
| | Nº hidden layers $\mathbf{f}_\theta$ | 2 |
| | Hidden dimension of $\mathbf{f}_\theta$ | 256 |
| | Nº hidden layers $g_\theta$ | 2 |
| | Nº hidden dimension $g_\theta$ | 128 |
| | Nº iterations GCN | 2 |
| | Nº Gaussian basis functions | 16 |
| | Hidden edge embedding dimension $D_{edge}$ | 32 |
| | Hidden node embedding dim. | 16 |
| | Activation function | SiLU |
| **Markov Chain Monte Carlo** | Nº walkers | 2048 |
| | Nº decorrelation steps | 20 |
| | Target acceptance prob. | 50% |
| **Variational pre-training** | Optimizer | KFAC |
| | Damping | $1 \times 10^{-3}$ |
| | Norm constraint | $3 \times 10^{-3}$ |
| | Batch size | 2048 |
| | Initial learning rate $lr_0$ | 0.1 |
| | Learning rate decay | $lr(t) = lr_0(1+t/6000)^{-1}$ |
| | Optimization steps | 64,000-512,000 |
| **Changes for fine-tuning** | Learning rate decay | $lr(t) = lr_0(7+t/6000)^{-1}$ |
| | Optimization steps | 0-32,000 |

no graph convolutional network. The medium-sized model uses one hidden layer of width 64 for $g_\theta$ and 128 for $\mathbf{f}_\theta$, and one iteration of the graph convolutional network. The small, medium and large models respectively have 0.8 mio, 1.2 mio. and 2.0 mio parameters.

## Data availability
All geometry- and energy-data is available on GitHub under https://github.com/mdsunivie/deeperwin. Model weights are available on figshare under https://doi.org/10.6084/m9.figshare.23585358.v1[43]. Source data are provided with this paper.

## Code availability
All code is available on GitHub under https://github.com/mdsunivie/deeperwin and Zenodo (https://doi.org/10.5281/zenodo.10081846)[44].

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

## Acknowledgements

We gratefully acknowledge financial support from the following grants: Austrian Science Fund FWF Project I 3403 (P.G.), WWTF-ICT19-041 (L.G.). The computational results have been achieved using the Vienna Scientific Cluster (VSC). The funders had no role in study design, data collection and analysis, decision to publish or preparation of the manuscript. Additionally, we thank Nicholas Gao for providing his results and data, Ruard van Workum for initial work on the python implementation for multi-compound optimization and Jan Hermann for fruitful discussions.

## Author contributions

M.S., L.G., and P.G. conceived the overall idea. M.S. conceived and implemented the ansatz, built the dataset and designed the experiments. L.G. gave input on the ansatz and worked on implementation. M.S. and L.G. performed the experiments. M.S. and L.G. wrote the manuscript with input, supervision and funding from P.G.

## Competing interests

The authors declare no competing interests.
