## [Peer Review File · Nature Communications]

Towards a Transferable Fermionic Neural Wavefunction for MoleculesREVIEWER COMMENTS

Reviewer #1 (Remarks to the Author):

The work by Grohs et al. achieves an important milestone in quantum chemistry by creating a neural network approach capable of learning the wavefunction of multiple different compounds and configurations of the same compound.

I have the following, mostly technical, comments that should be addressed in the revised manuscript:

1. It is currently unclear to a potential user of the method, how slow it is and whether it can be used in practical simulations. The authors only provide very uninformative statements that their method is efficient. What is the scaling of computational time with the number of electrons/atoms? What is the wall clock time required for a typical simulation on typical hardware (CPU/GPU)?
2. All notations should be clearly defined in the first place they are mentioned, e.g., most of the terms of eq. 1 and another non-labeled equation for h_{inu} are not explicitly defined in the main text but only later in the appendix.
3. The code (and key data) should be made available for review, otherwise, it is impossible to independently verify whether the results and conclusions hold.
4. What is the reference in Figure 5, to which their results are compared to?
5. Make it more clear in the main text and caption to Fig. 4 that DL-VMC was used as reference.
6. What are the values of parameters a , b , c for CBS extrapolation and how they were obtained? Taken from literature or fitted?
7. Please elaborate in more detail with the numerical comparison on the statement: 'we find that our ansatz currently does not fully match the accuracy of state-of-the-art single-geometry DL-VMC ansatz'?
8. Small typos:
 - 8.1. achieves lower absolute energies → achieves lower errors in absolute energies?
 - 8.2. Wave-package → wave-packet
 - 8.3. It mainly attributes → it is mainly attributed
 - 8.4. Promised for perform → promising for performing
 - 8.5. Leading to produce → leading to
 - 8.6. Dominate photoproduct → dominating photoproduct

Reviewer #2 (Remarks to the Author):

The authors propose a set of new techniques that push the boundaries of current efforts to solve the electronic Schrödinger equation using neural network. The key idea is to build locality into the backflow transformation and re-use a pretrained mapping of the backflow parameters.

There is an inherent assumption of locality in this approach which is reminiscent of ideas e.g. in machine-learned interatomic potentials and hamiltonians or in localised reduced basis functions for homogenising PDEs. In those contexts there is strong justification for such localisation but is this the case here? It is not discussed in the manuscript. How general is the approach? More discussion on this point would be appreciated.

Regardless of this open question, it is an interesting attempt that brings new ideas into the field of DNN wave functions. The results appear impressive, if somewhat limited in scope. The promise of a "foundation model" is visible from the paper but maybe still a bit of a stretch. This is clearly acknowledged in the title ("towards") and the conclusion. Minor comments follow.

[1] p. 3, Eq 1-3 : please specify your symbols. Most can be guessed but only if one knows the field. Especially $\{r\}$ is very confusing if one doesn't already know how backflow wave functions are normally constructed.

[2] p. 4, the new architecture: this was hard to read for several reasons:

(i) please give it equation numbers so one can reference this important block of equations.

(ii) why has the notation changed. Please make it consistent with (1-3).

(iii) the functions g^s, f^a should be written $g^{\text{rm } s}, f^{\text{rm } a}$ to indicate that a, s are labels and not variables.

(iv) symmetric and anti-symmetric is confusing because of the role of symmetry under permutation of electrons. Would it be possible to call them even and odd instead?

(v) I found the notation $f_{(d \nu)}$ confusing. I would strongly prefer $f_{\{\theta, d \nu\}(\dots)}$ or some alternatives. Why not $f_{\theta^{\{\text{rm } a\}, d}(c) \cdot h_i}$. Clearly a matter of taste but worth a brief discussion?

(vi) full stop missing.

[3] p. 4: "transferable atomic orbitals (TAOs)" - I've never heard that term and couldn't find it via google either. Please give a reference. The detailed description later in the paper is ok but I found the results section couldn't be read independently.

[4] It feels like there are many implicit assumptions built into this architecture. Can you discuss what they are? Related (but different) is the question whether the architecture in principle even complete/systematic/universal/ whatever you like to call it?

[5] p. 4: "the number of parameters is independent of system size" : that is formally true, but it is of course incorrect in the sense that the accuracy will be reduced with increasing system size when you keep the number of parameters the same. Once you fine-tune you presumably free the parameters and then they will scale with system size? This should be qualified here I think.

[6] p. 5 near the end : "sufficiently expressive to model ground state wfs to high accuracy" - this is a very generic statement, please qualify it. What kinds of systems, under which conditions? This is related to [4].

[7] p. 6 : "zero-shot accuracy" - I think this term is never defined.

[8] Figure 3 : completely unreadable, errors should almost always be on a log-scale

[9] p. 7, top second column: why are these model systems significant? Are they in fact? Or just the simplest possible systems to practice with?

[10] p.7 : "...results against the number of MCMC-samples ... linked to computational cost" : this is not at all obvious to me in this setting due to the cost of the Forward-laplacian - it relies on the assumption that you still need FAR MORE sample steps than gradient steps. Is this still the case in your setting?

[11] Figure 4

(i) Sub-plots (a-left) and (a-center) appear to show NEGATIVE ERRORS, impressive... maybe a log-scale would elucidate this again?

(ii) sub-plot c - I found it a bit harsh to call GLOBE at 10 deg a spurious minimum. It is clearly a fluctuation on the scale of the approximation error, maybe this can be said a bit differently.

[12] p. 9, Fig 5 : I thought that the first version of your new method has value independent of the idea to push it towards a foundation model. Maybe it could be included in this experiment?

[13] p. 10 : "...a general wavefunction..." : again a bit too (much too) generic, needs to be rephrased please.

[14] p.11 : The closing statement "While we are constrained..." is strangely defensive and should be rephrased positively.

[15] p. 13, left, first display : a superscript is missing, it should be $c_{|k|}$.

Reviewer #3 (Remarks to the Author):

This is a very interesting and timely manuscript, in which the authors introduce a general-purpose model neural-network based wave function that can describe several systems.

The main idea of finding general chemical models for neural wave functions already has some previous history. Most notably, Gao et al introduced a conceptually similar approach in previous works ([15,16,25]). Also, there is much related work by Zhang and Di Venira that has not been cited Phys. Rev. B 107, 075147 (2023), and should be mentioned in the text.

However, despite the fact this is not the first paper proposing the main conceptual step of a general

model+fine tuning, this is the one that seems to get closer to reasonable (see below) accuracy on benchmark cases. Especially interesting is the main ingredient behind the success is the introduction of generalized descriptors for the orbitals.

For this reason I believe the work is potentially worth publishing in Nature Comm., but there are several strong limitations of the method that should be spelled out more clearly and that need further numerical analysis.

1. In my view, the main limitation of this approach is that it cannot compete, even after fine tuning, with a specialized simulation of a given system using a neural-network ansatz specific for the number of electrons chosen. Also, because of the intrinsic locality of the orbitals, I suspect that this approach fails spectacularly at describing non-local correlations and critical systems. While this might not be practically relevant for some applications in chemistry, it is however quite crucial for condensed matter, and makes these models unusable in that context. A comment on these aspects would be needed.

2. For the Hydrogen chain, benchmarks energies are reported for the MRCI+Q (for which I couldn't find a reference in the text, but it might be my fault, maybe there is one?). Would it be possible to compare to state of the art results obtained with many methods, as done in Phys. Rev. X 7, 031059 (2017) ? It would be important to understand, after fine tuning, what is the energy differences obtained compared to state of the art.

3. In this sense, I do not find very comforting the following statement "While our approach yields Hartree-Fock-like energies for very long chains, GLOBE+ FermiNet and GLOBE+Moon yield results that are outperformed even by assuming a chain of non-interacting H-atoms", which seems to suggest that the model presented here barely recovers HF when generalizing to large chains. If this is the case, I am not sure this work (as much as the previous ones) are already at a stage where they are usable and improve on any (even cheaper) correlated method.

3. It would be useful if the authors obtained a potential energy surface for a single diatomic molecule where state-of-the-art results obtained with PsiFormer or similar are available, so to make a direct comparison.

Reviewer #4 (Remarks to the Author):

This article is concerned with the fundamental problem of computing accurate electronic states of molecules in a manner that alleviates the curse of dimension. The general approach follows that of Herrmann et al and Pfau et al: combine a quantum chemistry method of choice (variational Monte Carlo with Jastrow factor) with the backflow ansatz represented by a suitably trained deep neural net. But there is a significant advance: the weights in the backflow transformation are optimized on references systems with different electron number and varying geometries, and in the end are found to be transferrable to larger systems with only a small amount of additional tuning. This is loosely similar to

the ground-breaking decades-old practice of designing exchange-correlation functionals in density functional theory, whose parameters are optimized on ensembles of reference systems. Except that here the goal is to achieve (compared to DFT) much higher accuracy but for smaller systems, of a few dozen electrons. The method is tested on some small hydrocarbons and, more notably, on a chain of up to 28 hydrogen atoms, where it performs favourably against some reference methods.

The overall idea to achieve an accurate ansatz with chemically transferrable weights is novel and significant. It brings a version of the empirical principle of "chemical transferrability" of properties of molecular subgroups into the age of accurate scientific computing. This is a very nice paper which should be of interest across the fields of physics, chemistry, and molecular science, and could warrant publication in Nature Communications if suitably improved along the lines of the detailed comments below.

Detailed comments:

1) The H_n results should not just be compared to multi-reference CI but also to coupled cluster and more recent methods like tensor network methods, see e.g. Motta et al, PRX 7, 031059.

2) Figure 2: The large difference between the solid blue graph (not fine tuned) and the dashed blue graph (fine tuned) needs more explanation. Do I understand correctly that the transferrable atomic orbitals trained on 10 atoms only reduce the error of classical Hartree-Fock orbitals by a factor 2 for 15 atoms, and don't reduce it at all for 20 atoms? What, then, is the physical interpretation of the tuning, which seems to have a rather bigger role than its suggested name, 'fine-tuning'?

3) Figure 2: The difference between the dashed blue line (present work) and the reference should be indicated more precisely, it is somehow hidden by the chosen visualization. For researchers employing other methods to become interested in the new method, it is important that the absolute error, not divided by the number of atoms and not in cents of Hartrees but tenths of millihartrees, is given. Perhaps this can be done in a table. This serves to see how far the new method is from reaching 'chemical accuracy' (1 kcal/mole or 1 millihartree) for 30 electrons.

4) Can the discussion of the backflow ansatz in section 2.1 be a bit slower and more self-contained? What is the difference to Hermann et al and Pfau et al? Which combinations of electron positions $r_1, \dots, r_{n_{el}}$ are actually used - really full combinations?

5) Notation: eq. (1) introduces h_i , but not its components $h_{i\nu}$, but the latter are used in eq. (2).

6) Figure 4, middle row, last two panels: the convergence does not seem to be monotone with the number of samples, what is the explanation? Or, asked differently, is there a recommended choice of the number of samples for fine-tuning?

7) Section 4.3. The separation between electron embedding and orbital descriptor is certainly interesting but not fully unique, for instance it seems to the referee that one could unitarily transform the h_i 's and the f^a_θ 's without affecting the final wavefunction. Please comment.

Response to the reviewers' comments on "Towards a Foundation Model for Neural Network Wavefunctions"

Overall comments to all reviewers	1
Comments for Reviewer #1	1
Comments for Reviewer #2	3
Comments for Reviewer #3	7
Comments for Reviewer #4	9

Overall comments to all reviewers

We want to thank all reviewers for their thoughtful and thorough comments. We are delighted by the overall positive feedback on our work, calling it, "*an important milestone in quantum Chemistry*" (rev. 1), and "*novel and significant*" (rev. 4). We are glad to read that our conceptually new approach is appreciated as being "*an interesting attempt that brings new ideas into the field of DNN wave functions*" (rev. 2), and that our experiments "*seem to get closer to reasonable (see below) accuracy on benchmark cases. [...] potentially worth publishing in Nature Comm.*" (rev. 3).

We have addressed all your comments in the manuscript and individually below (reproducing the reviewer comments in black, and our answers in blue), but want to highlight some major improvements of the paper, which address suggestions common to several reviews:

- **Additional experiments and benchmarks:** We added 2 explicit quantitative comparisons of our method against other high-accuracy methods. For the Hydrogen-chain experiments we now compare our results against the energies obtained by Motta et al., and for the final base-model we evaluate its accuracy on the dissociation of a carbon dimer.
- **Clearer description of method:** We have improved the description of our method and improved the consistency of notation – both within the manuscript, as well as against established conventions. We have ensured that the methods section of this work is self contained, and have attempted to give intuition on design choices.
- **Code and Data availability:** All data, code and model-weights have been made publicly available.

Comments for Reviewer #1

The work by Grohs et al. achieves an important milestone in quantum chemistry by creating a neural network approach capable of learning the wavefunction of multiple different compounds and configurations of the same compound.

I have the following, mostly technical, comments that should be addressed in the revised manuscript:

1. It is currently unclear to a potential user of the method, how slow it is and whether it can be used in practical simulations. The authors only provide very uninformative statements that their method is efficient. What is the scaling of computational time with the number of electrons/atoms? What is the wall clock time required for a typical simulation on typical hardware (CPU/GPU)?

We added typical runtimes of our method (~30 min for zero-shot evaluation of a typical molecule in the test set with 3 heavy atoms) in the discussion and highlighted the $O(n_{el}^4)$ scaling of the per-iteration-cost.

2. All notations should be clearly defined in the first place they are mentioned, e.g., most of the terms of eq. 1 and another non-labeled equation for h_{inu} are not explicitly defined in the main text but only later in the appendix.

We added a dedicated notation block at the beginning of this section.

3. The code (and key data) should be made available for review, otherwise, it is impossible to independently verify whether the results and conclusions hold.

All code, geometries, and energies are now available on github. Model weights are available on figshare, and we have adjusted the code- and data-availability section accordingly.

4. What is the reference in Figure 5, to which their results are compared to?

Reference for Fig. 5 and 6 is CCSD(T)-CBS. In the revision we changed the y-label from “mean energy error” to “ $E - E_{CCSD(T)}$ ” to make this immediately obvious.

5. Make it more clear in the main text and caption to Fig. 4 that DL-VMC was used as reference.

We now clearly state the reference method in the caption.

6. What are the values of parameters a , b , c for CBS extrapolation and how they were obtained? Taken from literature or fitted?

We fitted the 5 parameters a , b , c , E_{HF_CBS} and E_{corr_CBS} . The values of a, b, c have different values for every single system (just like the CBS-energy and correlation energy).

We therefore did not take the values from literature, but only the functional form and cited it as Pfau et al. 2020.

7. Please elaborate in more detail with the numerical comparison on the statement: ‘we find that our ansatz currently does not fully match the accuracy of state-of-the-art single-geometry DL-VMC ansätze’?

We added a comparison of absolute energies for various DL-VMC ansätze (PsiFormer, PauliNet and its improved variants, our approach) in the appendix and find that on the example of NH₃, absolute energies between a fully trained PsiFormer and our fully trained network are ~1 mHa.

In addition we also include in the main text an experiment where we compare fine-tuning our pre-trained model against fully trained PsiFormer calculations (with 200x more compute). We find that in this setting our absolute energies are up to 20 mHa higher than PsiFormer, but still substantially better than CCSD(T) and FCIQMC.

8. Small typos:

8.1. achieves lower absolute energies → achieves lower errors in absolute energies?

8.2. Wave-package → wave-packet

8.3. It mainly attributes → it is mainly attributed

8.4. Promised for perform → promising for performing

8.5. Leading to produce → leading to

8.6. Dominate photoproduct → dominating photoproduct

8.1 is indeed correct, but we added half a sentence to clarify: Since both methods are variational, lower energies automatically also yield lower energy errors.

Regarding the phrases 8.2 - 8.6, we cannot find them anywhere in our manuscript and assume that the reviewer accidentally referred to another paper under review.

Comments for Reviewer #2

The authors propose a set of new techniques that push the boundaries of current efforts to solve the electronic Schrödinger equation using neural network. The key idea is to build locality into the backflow transformation and re-use a pretrained mapping of the backflow parameters.

There is an inherent assumption of locality in this approach which is reminiscent of ideas e.g. in machine-learned interatomic potentials and hamiltonians or in localised reduced basis functions for homogenising PDEs. In those contexts there is strong justification for such localisation but is this the case here? It is not discussed in the manuscript. How general is the approach? More discussion on this point would be appreciated.

We have added detail and motivation in both the section on locality in the main text, as well as the section on the localization of the Hartree-Fock orbitals.

We have no strict proof that local orbitals lead to the exact wavefunction, but several pieces of evidence suggest that locality is a useful prior: a) Many useful chemical concepts (such as core electrons, bonds, or chemical groups) are inherently local objects. b) As you point out, several observables such as energies or forces in machine-learned interatomic potentials can be well approximated by local descriptors, hinting at some locality of the underlying wavefunction leading to these local observables. c) Our approach builds on (approximate) locality and outperforms previous approaches on generalizing wavefunctions.

Regardless of this open question, it is an interesting attempt that brings new ideas into the field of DNN wave functions. The results appear impressive, if somewhat limited in scope. The promise of a

"foundation model" is visible from the paper but maybe still a bit of a stretch. This is clearly acknowledged in the title ("towards") and the conclusion. Minor comments follow.

[1] p. 3, Eq 1-3 : please specify your symbols. Most can be guessed but only if one knows the field. Especially $\{r\}$ is very confusing if one doesn't already know how backflow wave functions are normally constructed.

We included a notation block at the beginning of the result section, clarifying all notations including $\{r\}$.

[2] p. 4, the new architecture: this was hard to read for several reasons:

(i) please give it equation numbers so one can reference this important block of equations.

We added equation numbers to every equation.

(ii) why has the notation changed. Please make it consistent with (1-3).

We made the notation consistent with (1-3), removed the dependence on the embedding dimension in eq. 4 and introduced a dot product.

(iii) the functions g^a , f^a should be written g^{a} , f^{a} to indicate that a, s are labels and not variables.

Done.

(iv) symmetric and anti-symmetric is confusing because of the role of symmetry under permutation of electrons. Would it be possible to call them even and odd instead?

Thanks for the comment, we changed to the less confusing nomenclature of even and odd.

(v) I found the notation $f_{\{d \nu\}}$ confusing. I would strongly prefer $f_{\{\theta, d\nu\}}(\dots)$ or some alternatives. Why not $f_{\{\theta^{\text{a}}, d\}(\text{c})} \cdot h_i$. Clearly a matter of taste but worth a brief discussion?

We followed the proposed idea, introduced a dot product as \langle , \rangle , switched to e and a respectively, and now denote all indices before the function argument (see also [2] iii).

(vi) full stop missing.

Fixed.

[3] p. 4: "transferable atomic orbitals (TAOs)" - I've never heard that term and couldn't find it via google either. Please give a reference. The detailed description later in the paper is ok but I found the results section couldn't be read independently.

The term "transferable atomic orbitals (TAOs)" is introduced in this manuscript as a naming scheme for the newly introduced orbitals, capable of transfer across systems of varying numbers of particles. We use this term to distinguish our approach from the standard backflow in the subsequent experiments. To clarify this, we changed the manuscript to "which we call in the following transferable atomic orbitals (TAOs)," (see end of Sec. 2.1.)

[4] It feels like there are many implicit assumptions built into this architecture. Can you discuss what they are? Related (but different) is the question whether the architecture in principle even complete/systematic/universal/ whatever you like to call it?

Section 2.2 (properties of our ansatz) discusses the properties enforced by our ansatz, and we have added discussion on the motivation/assumptions, in particular regarding locality. Furthermore we compare the universality of our ansatz with other approaches in the appendix, showing that our ansatz is in principle less expressive, but only by not being able to represent wavefunctions which depend on the (physically irrelevant) permutation of atoms.

[5] p. 4: "the number of parameters is independent of system size" : that is formally true, but it is of course incorrect in the sense that the accuracy will be reduced with increasing system size when you keep the number of parameters the same. Once you fine-tune you presumably free the parameters and then they will scale with system size? This should be qualified here I think.

As stated in the comment, the number of parameters of our neural network wavefunction ansatz is constant with varying size of particles. This was not the case in previous approaches, e.g. FermiNet and PauliNet, where the matrix F (see eq. 2) was dependent on the number of particles. For fine-tuning we don't "free the parameters" in the sense that we increase the number of parameters with the number of particles but use the exact same architecture/ ansatz and therefore same number of parameters.

We agree that in general for larger and therefore more complex systems a more expressive neural network wavefunction might be necessary to fully describe the true wavefunction. This can be seen in follow-up work from DeepMind, where they increased the total number of parameters to achieve higher accuracy. We therefore slightly adapted the wording and rephrased it to "**In principle** the number of parameters is independent of the system size".

[6] p. 5 near the end : "sufficiently expressive to model ground state wfs to high accuracy" - this is a very generic statement, please qualify it. What kinds of systems, under which conditions? This is related to [4].

We added in the appendix a section showing empirically on ammonia the expressivity of the model against other DL-VMC methods in particular high-accurate methods such as PsiFormer. Compared to other methods incorporating ab-initio orbitals we reach higher accuracy (4-12 mHa improvement), but slightly lower accuracy compared to PsiFormer (~ 1mHa difference).

[7] p. 6 : "zero-shot accuracy" - I think this term is never defined.

We added an explanation ("i.e. without fine-tuning") the first time the statement appears.

[8] Figure 3 : completely unreadable, errors should almost always be on a log-scale

Switched to a log-scale.

[9] p. 7, top second column: why are these model systems significant? Are they in fact? Or just the simplest possible systems to practice With?

All three systems are known to be “failure” cases for methods like Hartree Fock or CCSD(T), have therefore been studied in earlier work on Neural Network wavefunctions, and therefore lend themselves well for a systematic comparison. For the Hydrogen-chains the correlation energy contributes substantially to the ground-state energy, as discussed in Motta et al. Similarly for the automerization of Cyclobutadiene¹ and the 90° twisted geometry of Ethene² even high-accuracy methods like CCSD(T) struggle with the multi-reference character and significantly overestimate the energy barrier. In addition these systems have few electrons, making them tractable for accurate but potentially expensive methods.

We added a note in section 2.5, clarifying these reasons for system selection.

[10] p.7 : "...results against the number of MCMC-samples ... linked to computational cost" : this is not at all obvious to me in this setting due to the cost of the Forward-laplacian - it relies on the assumption that you still need FAR MORE sample steps than gradient steps. Is this still the case in your setting?

This was unfortunately poorly phrased in the original manuscript and has now been worded more clearly. The computational cost is indeed – as pointed out by you – dominated by the cost of the forward-laplacian for the estimation of the kinetic energy. Therefore the key metric for total cost is the total number of laplacian evaluations, or equivalently the total number of samples for which the energy/energy-gradient is evaluated. This is what we plot on the x-axis of our graphs and is simply given by number-of-optimization-steps times batch-size. We did not plot the total number of intermediate MCMC samples generated during the de-correlation steps, as suggested by your comment.

The reason why we do not simply use nr-of-optimization-steps as x-axis, is that different methods we compare against have chosen different batch-sizes (and thus computational cost per step), making this a potentially unfair comparison.

[11] Figure 4

(i) Sub-plots (a-left) and (a-center) appear to show NEGATIVE ERRORS, impressive... maybe a log-scale would elucidate this again?

In the original manuscript we had used the potentially misleading axis title “energy error”. In the updated manuscript we changed it to “E - E_ref” and “ $\Delta E - \Delta E_{ref}$ ”. Our absolute energy predictions are variational, meaning we are lower-bounded by the ground-state energy, i.e. lower (variational) energy predictions are always closer to the ground truth. If E-E_ref (which we misleadingly named energy error) is negative, it simply means that we predict energies that are more accurate than the reference method. For H10 and Ethene we used MRCI as reference method, for Cyclobutadiene we used FermiNet (since we have not found public MRCI results).

(ii) sub-plot c - I found it a bit harsh to call GLOBE at 10 deg a spurious minimum. It is clearly a fluctuation on the scale of the approximation error, maybe this can be said a bit differently.

We removed this overzealous statement regarding the spurious minimum.

[12] p. 9, Fig 5 : I thought that the first version of your new method

¹ *Chem. Rev.*, 112, 1, 182–243 (2012), Multireference Nature of Chemistry: The Coupled-Cluster View

² *Nat Comput Sci* 2, 331–341 (2022), Solving the electronic Schrödinger equation for multiple nuclear geometries with weight-sharing deep neural networks

has value independent of the idea to push it towards a foundation model.
Maybe it could be included in this experiment?

The 2 types of lines in Fig. 5 (solid vs. dashed) actually correspond to our new method with and without “foundation model” pre-training. The dashed line is the energy we obtain using our proposed architecture without variational pre-training.

A potential confusion is the fact that without pre-training, weights for the variational optimization are typically not initialized randomly, but by an initial supervised fitting of the wavefunction against Hartree-Fock reference orbitals. This procedure was introduced by FermiNet (Pfau et al.) and is also called pre-training.

To avoid this confusion, we change the legend of Fig. 5 to “Train from scratch” instead of “HF-pre-training” to avoid the confusion between the variational pre-training and the supervised Hartree-Fock-pretraining.

[13] p. 10 : "...a general wavefunction..." : again a bit too (much too) generic, needs to be rephrased please.

Clarified as “[...] a wavefunction, which is transferable across compounds [...]”

[14] p.11 : The closing statement "While we are constrained..." is strangely defensive and should be rephrased positively.

Removed this strangely defensive qualifier.

[15] p. 13, left, first display : a superscript is missing, it should be $c_{\mathbf{k}}$.

Thank you for the hint. We adapted the notation.

Comments for Reviewer #3

This is a very interesting and timely manuscript, in which the authors introduce a general-purpose model neural-network based wave function that can describe several systems.

The main idea of finding general chemical models for neural wave functions already has some previous history. Most notably, Gao et al introduced a conceptually similar approach in previous works ([15,16,25]). Also, there is much related work by Zhang and Di Ventura that has not been cited Phys. Rev. B 107, 075147 (2023), and should be mentioned in the text.

Thank you for pointing out the paper by Zhang et al., which we had not been aware of. We now explicitly mention their work in the introductory section.

Their paper does indeed share the main idea of pre-training a general purpose model. However, as you point out, there are marked differences, such as the type of system (2nd quantization model Hamiltonians vs. 1st quantization molecules), the model architecture (transformer vs. Graph Neural Network), and our architectural innovation of using mean-field orbitals as descriptors for the orbitals.

However, despite the fact this is not the first paper proposing the main

conceptual step of a general model+fine tuning, this is the one that seems to get closer to reasonable (see below) accuracy on benchmark cases. Especially interesting is the main ingredient behind the success is the introduction of generalized descriptors for the orbitals.

For this reason I believe the work is potentially worth publishing in Nature Comm., but there are several strong limitations of the method that should be spelled out more clearly and that need further numerical analysis.

1. In my view, the main limitation of this approach is that it cannot compete, even after fine tuning, with a specialized simulation of a given system using a neural-network ansatz specific for the number of electrons chosen. Also, because of the intrinsic locality of the orbitals, I suspect that this approach fails spectacularly at describing non-local correlations and critical systems. While this might not be practically relevant for some applications in chemistry, it is however quite crucial for condensed matter, and makes these models unusable in that context. A comment on these aspects would be needed.

Yes, optimizing a wavefunction specialized to a particular system for many optimization steps, will yield higher accuracy than using our pre-trained wavefunction (albeit at orders of magnitude larger computational cost). We had originally addressed this limitation in the discussion section, but have now made it much more explicit by: a) including the binding curve of a carbon-dimer as a prototypical case and comparing against specialized PsiFormer calculations b) comparing our H-chain results against other specialized approaches from the work by Motta et al.

We find that our results show lower accuracy than PsiFormer, but comparable (or superior) accuracy to CCSD(T), FCI-QMC, NEVPT2, and DMET.

It could be that our approach “fails spectacularly at describing non-local correlations”, or that these models are “unusable in that context”, but these systems are beyond the scope of this work, which focuses exclusively on non-periodic, gas-phase molecules. We note however, that we obtain promising accuracy on the thermodynamic limit of the H-chains, and that the current approach in principle correlates all electrons among each other (via the electron embedding h_i). It is therefore not clear to us whether this combination of globally correlated electron embeddings and local orbitals is fundamentally insufficient.

We believe this to be an exciting direction for further developments and would be grateful if the reviewer could recommend a (set of) small molecule(s) (ideally ≤ 40 electrons), which exhibits strong non-local correlations to use as a test system for future research in a follow-up paper.

2. For the Hydrogen chain, benchmarks energies are reported for the MRCI+Q (for which I couldn't find a reference in the text, but it might be my fault, maybe there is one?). Would it be possible to compare to state of the art results obtained with many methods, as done in Phys. Rev. X 7, 031059 (2017) ? It would be important to understand, after fine tuning, what is the energy differences obtained compared to state of the art.

We added a comparison against the methods from Motta et al. (PRX 2017), for the two cases present in the reference: the H10 chain and the thermodynamic limit. We find that after fine-tuning

our method outperforms most methods of the benchmark, with lower energies only achieved by AFQMC and RCCSD(T).

3. In this sense, I do not find very comforting the following statement "While our approach yields Hartree-Fock-like energies for very long chains, GLOBE+ FermiNet and GLOBE+Moon yield results that are outperformed even by assuming a chain of non-interacting H-atoms", which seems to suggest that the model presented here barely recovers HF when generalizing to large chains. If this is the case, I am not sure this work (as much as the previous ones) are already at a stage where they are usable and improve on any (even cheaper) correlated method.

It is true that for extrapolations far beyond the training regime (e.g. H10 to H28) our zero-shot results "only" recover the Hartree-Fock energy. This is still a feat that was not achieved by prior work, and is non-trivial, since we do not just learn an energy correction, but learn the full wavefunction and thus implicitly the corresponding total energy.

More importantly however, our method recovers 94% of correlation energy after just 500 fine-tuning steps (and 98% after 4000 fine-tuning steps), placing it among the most accurate correlated methods.

Whether this is already at a stage where it is usable for practitioners, will depend on the specific application, but we believe that our work could be of wide interest to both method developers as well as more applied practitioners.

3. It would be useful if the authors obtained a potential energy surface for a single diatomic molecule where state-of-the-art results obtained with PsiFormer or similar are available, so to make a direct comparison.

We ran a new experiment on the carbon-dimer PES, using the base-model pre-trained in Sec. 2.6 and compared it against CCSD(T), FCIQMC and PsiFormer (Fig. 6). Similar to the H-chain we find unsatisfactory accuracy in the zero-shot regime, but high accuracy after few pre-training steps (e.g. outperforming CCSD(T) in terms of accuracy of relative energies).

Comments for Reviewer #4

This article is concerned with the fundamental problem of computing accurate electronic states of molecules in a manner that alleviates the curse of dimension. The general approach follows that of Herrmann et al and Pfau et al: combine a quantum chemistry method of choice (variational Monte Carlo with Jastrow factor) with the backflow ansatz represented by a suitably trained deep neural net.

But there is a significant advance: the weights in the backflow transformation are optimized on reference systems with different electron number and varying geometries, and in the end are found to be transferrable to larger systems with only a small amount of additional tuning. This is loosely similar to the ground-breaking decades-old practice of designing exchange-correlation functionals in density

functional theory, whose parameters are optimized on ensembles of reference systems. Except that here the goal is to achieve (compared to DFT) much higher accuracy but for smaller systems, of a few dozen electrons. The method is tested on some small hydrocarbons and, more notably, on a chain of up to 28 hydrogen atoms, where it performs favourably against some reference methods.

The overall idea to achieve an accurate ansatz with chemically transferrable weights is novel and significant. It brings a version of the empirical principle of "chemical transferrability" of properties of molecular subgroups into the age of accurate scientific computing. This is a very nice paper which should be of interest across the fields of physics, chemistry, and molecular science, and could warrant publication in Nature Communications if suitably improved along the lines of the detailed comments below.

Detailed comments:

1) The H_n results should not just be compared to multi-reference CI but also to coupled cluster and more recent methods like tensor network methods, see e.g. Motta et al, PRX 7, 031059.

We added a thorough comparison of our approach against Motta et al, including extrapolations to the thermodynamic limit (TDL) and a comparison against all methods in Motta et al. for which TDL-energies are available.

2) Figure 2: The large difference between the solid blue graph (not fine tuned) and the dashed blue graph (fine tuned) needs more explanation. Do I understand correctly that the transferrable atomic orbitals trained on 10 atoms only reduce the error of classical Hartree-Fock orbitals by a factor 2 for 15 atoms, and don't reduce it at all for 20 atoms? What, then, is the physical interpretation of the tuning, which seems to have a rather bigger role than its suggested name, 'fine-tuning'?

By fine-tuning we mean taking an ansatz which was pre-trained on different systems (e.g. short H-chains) and optimizing it for a specific new task (e.g. long H-chains). This is consistent with the usage in the wider deep-learning literature, for example in the field of natural language processing (NLP), where models are typically pre-trained on internet-scale data and subsequently fine-tuned on a specific task (e.g. question-answering) using few additional optimization steps on this specific problem.

Similar to NLP, we find that pre-trained models transfer decently to new systems (i.e. we reach Hartree-Fock accuracy), but that much higher accuracy can be obtained by performing a few additional optimization steps. The exact physical interpretation of this fine-tuning (i.e. which previously unseen physics is being learned at this stage), is beyond the scope of our work. We speculate that it is in particular long-range interactions – which are not present in the short chains used for pre-training – that are being learned at this stage.

We also stress that our ansatz models the full wavefunction and the full energy and thus does not differentiate explicitly between Hartree-Fock energy and correlation energy. Even recovering the Hartree-Fock energy is not a completely trivial task for neural network wavefunctions, and prior work has failed to do so. To our knowledge our results are the first neural network ansatz which

obtains a physically reasonable (i.e. Hartree-Fock-like) energy without any fine-tuning, and high-accuracy energies after few additional optimization/fine-tuning steps.

3) Figure 2: The difference between the dashed blue line (present work) and the reference should be indicated more precisely, it is somehow hidden by the chosen visualization. For researchers employing other methods to become interested in the new method, it is important that the absolute error, not divided by the number of atoms and not in cents of Hartrees but tenths of millihartrees, is given. Perhaps this can be done in a table. This serves to see how far the new method is from reaching 'chemical accuracy' (1 kcal/mole or 1 millihartree) for 30 electrons.

We amended Fig. 2 to show a detailed comparison of our dashed blue line, against several reference methods presented in Motta et al.

We depict the errors both for 10 atoms, as well as in the thermodynamic limit (which are the 2 settings for which reference calculations are available). We find that our method reaches ~ 2 mHa error/atom after 500 fine-tuning steps and 0.5 mHa error/atom after 4k fine-tuning steps, being on par with many other accurate many-body-methods evaluated in Motta et al.

4) Can the discussion of the backflow ansatz in section 2.1 be a bit slower and more self-contained? What is the difference to Hermann et al and Pfau et al? Which combinations of electron positions $r_1, \dots, r_{\{n_{el}\}}$ are actually used - really full combinations?

We added a paragraph in section 2.1. highlighting the differences to Hermann et al and Pfau et al, in particular the differences in the envelope function and the generation of the high-dimensional electron embedding.

Indeed our embedding (as well as the embeddings of all modern neural-network wavefunctions) considers all combinations of electrons, i.e. each electron is sharing information with all other electrons (albeit with a learnable distance dependence). Since the embedding architecture used in this manuscript was proposed and discussed in an earlier work, we do not include a detailed derivation of computing the embedding vectors per electron, but give appropriate citations.

We also note that the approach proposed in this work can in principle work with any embedding network that yields feature vectors for each electron.

5) Notation: eq. (1) introduces h_i , but not its components $h_{i\nu}$, but the latter are used in eq. (2).

We changed the notation to be more consistent and replace the notation $h_{i\nu}$ with a dot product for eq. 2 and eq. 6 in the new manuscript.

6) Figure 4, middle row, last two panels: the convergence does not seem to be monotone with the number of samples, what is the explanation? Or, asked differently, is there a recommended choice of the number of samples for fine-tuning?

We do not see evidence for an optimal number of fine-tuning steps. We believe that fine-tuning does systematically improve accuracy, and that the apparent increase in "relative energy error" is an artefact of the finite accuracy of the reference method or an artefact of Monte Carlo noise. For the updated manuscript we have made several changes to show this more clearly and avoid confusion: For the original manuscript we had used the y-label "relative energy error" which is descriptive, but somewhat misleading, since there are no exact reference methods for these

systems. In the revised manuscript we use the label $\Delta E - \Delta E_{\text{ref}}$, which also has the advantage of showing both positive and negative deviations. The apparent small increase in relative energy error for the C₂H₄ barrier during the fine-tuning is mostly explained by a sign change of the error: During optimization VMC initially yields a slightly higher estimation than the MRCI reference, than a very similar relative energy, and ultimately a slightly lower relative energy than MRCI. When plotting the absolute difference of the relative energy, this leads to an apparent uptick in relative energy error.

7) Section 4.3. The separation between electron embedding and orbital descriptor is certainly interesting but not fully unique, for instance it seems to the referee that one could unitarily transform the h_i 's and the f^a_θ 's without affecting the final wavefunction. Please Comment.

Yes, given that we form an inner product between h_i and f_k (by summing over ν), we could multiply h_i by an invertible matrix U and f_k by its inverse and still obtain the same result. The same holds for other neural network wavefunctions, with the only difference being how $f_{\{k \nu\}}$ is obtained: In our case we obtain $f_{\{k \nu\}}$ as a function of the Hartree-Fock orbital k , whereas other neural wavefunctions (e.g. FermiNet, PsiFormer, DeepErwin) directly use a trainable matrix. In both cases a redundant parameterization is avoided, by only adding a linear output layer to one of the two functions (either f or h). In our case (as well as in FermiNet, PsiFormer, etc), we do not apply a linear output layer to the orbital embeddings h_i and instead perform a tanh-activation as the final step. As you rightly point out, any final linear transformation of the electron embeddings h_i , would be redundant, because it can be absorbed in the orbital embedding f_k , for which we do use a final linear output layer.

REVIEWERS' COMMENTS

Reviewer #1 (Remarks to the Author):

The authors adequately answered all my comments. I have no further comments.

Reviewer #2 (Remarks to the Author):

I'd like to thank the authors for the serious engagement with my remarks. however, there are three remaining points I'd like to come back to. I feel not addressing this leaves some minor misleading information in the paper.

[0] Regarding the discussion on localisatin: thank you for adding this. I think it would also be appropriate to cite relevant literature where similar locality priors were employed successfully. For example MLIPs (Behler/Parinello 2007, Bartok/Csnayi et al 2010) and then later there is a huge amount of work for potentials, hamiltonians and many other applications from many different groups.

[5] discussion re number of parameters and system size: I appreciate the change but I still think it is incorrect to claim this and it should be clarified maybe as follows: ... while in principle independent of systems size ... to maintain uniform accuracy for systems of increasing size ... it may be necessary to scale the number of parameters with system size.

[10] Re discussion on computational bottleneck: Your change in the manuscript "against the number of samples used for the energy estimation during variational optimization, which is very closely linked to computational cost." is contradictory with your acknowledgement that the gradients are the actual bottleneck and # gradient evaluations therefore determines the computational cost. Please correct the manuscript or explain the confusion?

Reviewer #5 (Remarks to the Author):

I have revised the manuscript entitled "Towards a foundational model for neural wave functions". In the manuscript a novel variational neural network wave-function (NQS) is introduced, for the treatment of the electronic struc-ture in molecular many-body systems. In particular, the introduced variational Ansatz allows the simulation of variably sized molecules, without any modifica-tions to its functional

form. Benchmarks w.r.t. different state-of-the-art NQS Ansätze as well as quantum chemistry methods are included.

The previous 4 referees have requested additional clarifications, comparisons to other state-of-the-art methods, and changes in figures/equations. I have verified whether the requested changes have been taken into account and comment in detail below for each of the reports. In general, I find that the article has improved with the new revision, it has become more clear and unambiguous. The authors have properly answered to the questions of the other referees. I recommend this manuscript for publication, given that the remaining minor comments have been taken into account, as mentioned below.

However, one major remaining obstacle that has been confusing for the community is the title of the manuscript, which in my opinion is much too general and simply confusing. The title should better narrow down what the paper is about, for example: molecular systems, transferability (which would be a more accurate term than a foundational model in my opinion), fermionic neural wave functions, etc. Given that there is an entire field actively working on neural quantum states (or neural wave functions), including continuous degrees of freedom, discrete degrees of freedom, lattice systems, etc., the title reaches too far, way beyond its scope.

Referee 1

The major part of the revision work for Referee 1 consists of improving notation in the manuscript. Additionally a more detailed explanation of the architecture and a comparison to other state-of-the-art NQS methods was asked for.

While the notation has been improved compared to the former version (added a dedicated block explaining the notation), there are still some minor inconsistencies. For example it is said that "all vectors, matrices and tensors are denoted in bold letters". In Equation 1, the function h_9 is not bold, even though it has a vectorial output. Same for functions f_9 in Equations 4,5. Additionally, while most symbols now get explained, where they are first defined, I could not find an explicit mention and explanation of the function h_9 in Equation 1, even though this is clearly one of the main ingredients to the Ansatz. There is more explanation in Section 4.2. but not where the function is first mentioned. I would add at least a reference to Section 4.2. The same holds for the function g_9 (above Equation (4) only f_9 gets introduced).

The comparison of NH3 and C2 results to PsiFormer, PauliNet etc., presented in the appendix and the main text (Figure 6), helps to understand improvements/shortcomings of the architecture in this work compared to more established ones, and constitutes an improvement to the manuscript.

Referee 2

The notational and technical revisions of Referee 2 are very similar to the ones of Referee 1 and therefore the same comment as above applies.

Content-wise, the Referee asks for a more in-depth discussion on the assumptions and biases entering the variational Ansatz, with special emphasis on the role of locality. Additionally he requests a

theoretical argument on the theoretical expressiveness of the Ansatz in use, in particular regarding the constant parameter count of the Ansatz.

In Section 2.2. the properties and assumptions of the Ansatz are discussed and a dedicated paragraph on the role of locality has been added. In the appendix a mathematical analysis has been carried out, comparing their Ansatz to established ones (FermiNet, PauliNet) which are known to be universal. Even though this mathematical argument seems consistent and shows equivalence between all architectures, the question remains, why FermiNet seems to perform better than this architecture on the test cases NH3 and C2. It would be nice if the authors could comment on this. The assumptions made in the construction of their Ansatz are explained well.

Referee 3

The Referee requests the authors to comment on the restrictions of the Ansatz in application to non-molecular systems such as solids. He/she also asks for additional benchmarks w.r.t. state-of-the-art methods in quantum chemistry as well as other NQS architectures, to better understand the accuracy of the proposed method.

The authors acknowledge the fact, that their method's scope is not condensed matter systems but "non-periodic, gas phase molecules". However in the text I am missing an explicit mention of this, as was asked by the Referee. The authors added the benchmarks proposed by the Referee (Motta et al. for H-chains, PsiFormer for a potential energy surface of C2) and I find the comparisons satisfactory, in the sense that they show what can be expected from the proposed Ansatz compared to more established ones.

Referee 4

The Referee asks for a more detailed explanation of the Ansatz used in this work, including a comparison to established Ansatzes like FermiNet and PauliNet. Additionally the difference between fine-tuned/non-fine-tuned results should be explained more clearly and additional benchmarks of the H-chain results should be included to the manuscript. Furthermore the Referee asks for a clearer visualization of the results in Figure 2.

I find that the notation in the manuscript has been improved (however the same comments apply as for Referee 1,2). The explanation of the Ansatz has improved thanks to the notation changes and the comparison to state-of-the-art NQS (FermiNet and PauliNet) helps to put this new Ansatz into context of other NQS architectures. The H-chain benchmarks have been added to the manuscript and a clearer comparison has been added in the form of panel (b),(c) in Figure 2. However contrary to the demand of the Referee to present the total energies (not divided by number of particles), the energies are still presented per particle. I personally do not find this an obstacle for publication, though.

Response to the reviewers' comments on "Towards a Transferable Fermionic Neural Wavefunction for Molecules"

Comments for Reviewer #2:.....	1
Comments for Reviewer #5.....	2

In the following we reproduce the reviewer comments verbatim in black and add our response in blue.

Comments for Reviewer #2:

1. Regarding the discussion on localisation: thank you for adding this. I think it would also be appropriate to cite relevant literature where similar locality priors were employed successfully. For example MLIPs (Behler/Parinello 2007, Bartok/Csnayi et al 2010) and then later there is a huge amount of work for potentials, hamiltonians and many other applications from many different groups.

We appreciate this comment and now cite the usage of locality for Neural Network Potentials as suggested. Since locality is a very natural prior and has been used extensively in the past, it appears impossible to exhaustively cite usage of this concept in prior work beyond this specific application.

2. discussion re number of parameters and system size: I appreciate the change but I still think it is incorrect to claim this and it should be clarified maybe as follows: ... while in principle independent of systems size ... to maintain uniform accuracy for systems of increasing size ... it may be necessary to scale the number of parameters with system size.

We agree and added a clarifying statement based on your suggestions:

"While it might still be necessary to increase the parameter count to maintain uniform accuracy for systems of increasing size, TAOs have at least no explicit relationship between parameter count and system size."

3. Re discussion on computational bottleneck: Your change in the manuscript "against the number of samples used for the energy estimation during variational optimization, which is very closely linked to computational cost." is contradictory with your acknowledgement that the gradients are the actual bottleneck and # gradient evaluations therefore determines the computational cost. Please correct the manuscript or explain the confusion?

We try in the following to clear up any confusion once and for all.
The cost per optimization step is split into two parts:

- 1) **Drawing samples r from \log_{ψ} squared:** This is done using Markov Chain Monte Carlo and involves $\text{batch_size} \times \text{nr_of_intermediate_steps}$ many evaluations of the wavefunction.
- 2) **Computing the gradient of the energy.** The energy gradient is given as the covariance between the local energies in a batch and the gradient of the wavefunction wrt. to the parameters: $\text{grad } E = \text{cov}(E_{\text{local}}, \text{grad}_{\theta} \log_{\psi})$. This therefore involves:
 - a) A forward pass through the network to obtain \log_{ψ}
 - b) A backward pass through the network to obtain $\text{grad } \log_{\psi}$
 - c) The evaluation of the local energies, which involve $3 \times n_{\text{electron}}$ many passes through the network to compute the second derivative required for the kinetic energy.
 - d) Computing the gradient preconditioner of the optimizer (K-FAC)
 - e) Applying the preconditioner and updating the parametersThe computational cost of steps a-d (which make up almost the entirety of the runtime) all are proportional to the batch size.

The point made in our paper is simply that since the total computational cost in almost all steps (1, 2a-d) is proportional to the batch size (and different works have used different batch sizes), we do not compare the number of optimization steps, but the $\text{nr_of_optimization_steps} \times \text{batch_size}$.

This product ($\text{nr_of_optimization_steps} \times \text{batch_size}$) is exactly the total number of samples for which the local energy is being evaluated, referred to by us as “the number of samples used for the energy estimation during variational optimization”.

The reviewer thinks that there is a contradiction between the following two statements.

- I. “[...] number of samples used for the energy estimation during variational optimization, which is very closely linked to computational cost”
- II. “# gradient evaluations therefore determines the computational cost”

However, this is no contradiction, since the energy evaluation (2c) is part of the gradient evaluation (2), and accounts for 90%+ of the gradient's computational cost. For typical settings the energy computation (2c) is the vast majority of overall computational cost.

We hope this clarifies any remaining confusion.

Comments for Reviewer #5

However, one major remaining obstacle that has been confusing for the community is the title of the manuscript, which in my opinion is much too general and simply confusing. The title should better narrow down what the paper is about, for example: molecular systems, transferability (which would be a more accurate term than a foundational model in my opinion), fermionic neural wave functions, etc. Given that there is an entire field actively working on neural quantum states (or neural wave functions), including continuous degrees of freedom, discrete degrees of freedom, lattice systems, etc., the title reaches too far, way beyond its scope.

We agree with the reviewer and have adapted the title to the following, more specific title:

“Towards a Transferable Fermionic Neural Wavefunction for Molecules”.

While the notation has been improved compared to the former version (added a dedicated block explaining the notation), there are still some minor inconsistencies. For example it is said that “all vectors, matrices and tensors are denoted in bold letters”. In Equation 1, the function h_θ is not bold, even though it has a vectorial output. Same for functions f_θ in Equations 4,5.

Thank you for pointing out the flaw in our notation. Previously, we had the convention that any function (independent of scalar or vectorial output) is not bold. We adapted it and now every function with vectorial output is highlighted in bold.

Additionally, while most symbols now get explained, where they are first defined, I could not find an explicit mention and explanation of the function h_θ in Equation 1, even though this is clearly one of the main ingredients to the Ansatz. There is more explanation in Section 4.2. but not where the function is first mentioned. I would add at least a reference to Section 4.2. The same holds for the function g_θ (above Equation (4) only f_θ gets introduced).

Thank you for your comment. We explicitly now mention the trainable function g when we first introduce it and reference to the relevant part of the method section.

For the embedding (h_{θ}) we now explicitly mention the function in the paragraph following eq. 1. Previously, we only mentioned the output h and discussed how different approaches compute the function:

“Existing methods, as proposed in Pfau et al. [2] or Hermann et al. [1], differ in the way the embedding h_i and the envelope functions are built... “

or

“A popular choice for the embedding function are continuous convolutions [1, 5, 26] or a transformer-based attention mechanism [4].”

To highlight the correspondence to eq. 1 we rephrased to:

“A popular choice for the embedding function h_{θ} “.

Even though this mathematical argument seems consistent and shows equivalence between all architectures, the question remains, why FermiNet seems to perform better than this architecture on the test cases NH3 and C2. It would be nice if the authors could comment on this.

We unfortunately do not currently have a good understanding of this discrepancy. Ultimately the mathematical proof of universality is only a necessary condition, but not sufficient to obtain highly accurate solutions in practice. We now acknowledge this issue in the final paragraph of the supplementary information.

A similar case can be found when comparing the accuracy of FermiNet and PsiFormer. In principle both these neural network architectures should be sufficiently expressive to represent the ground-state eigenfunction but empirically PsiFormer achieves substantially lower energies on certain systems. We believe to observe a similar thing and assume it to be connected partially to optimization which could potentially be improved with more exhaustive hyperparameter tuning.

The authors acknowledge the fact, that their method's scope is not condensed matter systems but "non-periodic, gas phase molecules". However in the text I am missing an explicit mention of this, as was asked by the Referee.

We mention now in the main text that our method is for non-periodic, gas-phase molecules: "In this work we propose a novel neural network ansatz, which does not depend explicitly on the number of particles, allowing to optimize wavefunctions across multiple **non-periodic, gas-phase compounds** with multiple different geometric conformations."